# Estimation of OH in urban plumes using TROPOMI inferred $NO_2$/ CO

**Srijana Lama[1], Sander Houweling[1,2], K. Folkert Boersma[3,4], Ilse Aben[2,1], Hugo A. C. Denier van der Gon[5], Maarten C. Krol[3,6]**

[1]Vrije Universiteit, Department of Earth Sciences, Amsterdam, the Netherlands

[2]SRON Netherlands Institute for Space Research, Leiden, the Netherlands

[3]Wageningen University, Meteorology and Air Quality Group, Wageningen, the Netherlands

[4]Royal Netherlands Meteorological Institute, R&D Satellite Observations, de Bilt, the Netherlands

[5]TNO, Department of Climate, Air and Sustainability, Princetonlaan, the Netherlands

[6]Institute for Marine and Atmospheric Research Utrecht, Utrecht University, Utrecht, the Netherlands

*Correspondence to*: Srijana Lama (s.lama@vu.nl, sreejanalama@gmail.com)

**Abstract.**

A new method is presented for estimating urban hydroxyl radical (OH) concentrations using the downwind decay of the ratio of nitrogen dioxide over carbon monoxide column mixing ratios ($XNO_2$ / XCO) retrieved from the Tropospheric Monitoring Instrument (TROPOMI). The method makes use of plumes simulated by the Weather Research and Forecast model (WRF-CHEM) using passive tracer transport, instead of the encoded chemistry, in combination with auxiliary input variables such as Copernicus Atmospheric Monitoring Service (CAMS) OH, Emission Database for Global Atmospheric Research v4.3.2 (EDGAR) NOx and CO emissions, and National Center for Environmental Protection (NCEP) based meteorological data. $NO_2$ and CO mixing ratios from the CAMS reanalysis are used as initial and lateral boundary conditions. WRF overestimates $NO_2$ plumes close to the center of the city by 15 % to 30 % in summer and 40 % to 50 % in winter compared to TROPOMI observations over Riyadh. WRF simulated CO plumes differ by 10 % with TROPOMI in both seasons. The differences between WRF and TROPOMI are used to optimize the OH concentration, NOx, CO emissions and their backgrounds using a iterative least square method. To estimate OH, WRF is optimized using a) TROPOMI $XNO_2$/XCO,  b) TROPOMI derived $XNO_2$ only.

For summer, both the $NO_2$/CO ratio optimization and the $XNO_2$ optimization increase the OH prior from CAMS by $32 \pm 5.3$ % and $28.3 \pm 3.9$ % respectively. EDGAR NOx and CO emissions over Riyadh are increased by $42.1 \pm 8.4$ % and $101 \pm 21\%$, respectively, in summer. In winter, the optimization method doubles the CO emissions also, while increasing OH by $\sim 52 \pm 14$ % and reducing NOx emission by $15.5 \pm 4.1$ %.  TROPOMI derived OH concentrations and pre-existing Exponentially Modified Gaussian function fit (EMG) method differ by 10 % in summer and winter, confirming that urban OH concentrations

can be reliably estimated using the TROPOMI-observed $NO_2/CO$ ratio. WRF optimization method can be applied to single TROPOMI overpass, allowing to analysis day to day variability in OH, NOx and CO emission.

## 1. Introduction

The rapidly growing urbanization has led to an increase in the number of big cities globally. More than 55 % of the global population resides in cities and this fraction is projected to increase to 68 % in 2050 (United Nations, 2018). The associated
rise in consumption of energy and materials leads to severe air pollution that is estimated to have caused premature death of 4 to 9 million people globally in 2015 (Sicard et al., 2021; Pascal et al., 2013; Burnett et al., 2018). Air pollution control measures and the application of cleaner technology have reduced the $NO_2$ concentrations in developed cities such as Los Angeles and Paris by 1.5 % to 3.0 % $yr^{-1}$ between 1996 to 2017 (Georgoulias et al., 2019). The CO emission is reduced by 28.8 % to 60.7 % in these cities in the period 2000 to 2008 (Dekker et al., 2017). In developing cities such as Tehran and Baghdad, however,
$NO_2$ concentrations have increased by 8.6 % $yr^{-1}$ and 16.9 % $yr^{-1}$ between 1996 to 2017 (Georgoulias et al., 2019). The CO emission increased by 15 % in New Delhi in the period 2000 to 2008 (Dekker et al., 2017). As a consequence, air pollution monitoring and mitigation in developing cities is becoming an increasingly important priority.

Nowadays, urban air pollution can be studied using a combination of ground-based measurement networks and satellite observations (Sannigrahi et al., 2021; Ialongo et al., 2020). Satellite observations have helped to investigate urban air pollution,
particularly in cities without a ground-based monitoring network (Beirle et al., 2019; Borsdorff et al., 2019). In past decades, improvements in the quality and spatial resolution of satellite measurements have allowed the detection of trends in air pollutants and the quantification of urban emissions (Lorente et al., 2019; Verstraeten et al., 2018; Wennberg et al., 2018). Several studies have focused on NOx, using $NO_2$ observations from the SCanning Imaging Absorption spectroMeter for Atmospheric CartograpHY (SCIAMACHY) , the Ozone Monitoring Instrument (OMI) and TROPOMI (Ding et al., 2017;
Lorente et al., 2019). At the resolution and sensitivity of TROPOMI, urban $NO_2$ enhancements can be detected readily, even in single satellite overpass. OMI derived $NO_2$ data have been used to quantify NOx emissions, as well as the urban lifetime of $NO_2$, as demonstrated by Beirle et al. (2011) using the Exponentially Modified Gaussian function fit (EMG) method.

In the EMG method, the satellite observed exponential decay of $NO_2$ downwind of the city centre is used to quantify the first order loss of $NO_2$, which is used to quantify the hydroxyl radical (OH) neglecting other NOx removal pathways. Liu et al.
(2016) modified the EMG method for application to complex emission patterns. The quantification of CO emissions from cities is more complicated compared with $NO_2$ because of its longer lifetime, and the related importance of CO sources from the surroundings of cities. Nevertheless, a few studies have demonstrated the feasibility of quantifying relative changes in urban CO emission, using Measurement of Pollution in the Troposphere (MOPPIT), Infrared Atmospheric Sounding Interferometer (IASI), Atmospheric Infrared Sounder (AIRS), and TROPOMI observations (Borsdorff et al., 2019; Dekker et
al., 2017; Pommier et al., 2013).

In recent years, methods have been developed that combine satellite measurements of different trace gases, for example the combined use of $NO_2$ and CO, to obtain specific information about pollutant sources (Lama et al., 2020, Hakkarainen et al., 2015; Miyazaki et al., 2017; Reuter et al., 2019; S. Silva & Arellano, 2017 ). The emission factors of CO and $NO_x$ from fuel combustion are uncertain and vary strongly with the combustion efficiency (Flagan and Seinfeld, 1988). The satellite observed

$\Delta NO_2/\Delta CO$ ratio is particularly sensitive to this fuel burning efficiency, as demonstrated by Lama et al., (2020) and can be used to evaluate emission inventories. However, another important uncertainty arises from the removal of $NO_2$ by OH. OH is an important oxidant in the atmosphere, which determines the lifetime of trace gases such as CO, NOx, sulphur dioxide ($SO_2$) and volatile organic compound (VOCs) (Monks et al., 2009) . OH plays the important role in atmospheric chemistry on scales ranging from urban air pollution to the global residence times of greenhouse gases. The direct measurement of OH is possible

using spectroscopic methods, but the spatial representativeness of the data is limited due to its short lifetime (de Gouw et al., 2019). OH estimates from global Chemical Transport Models (CTM's) has an uncertainty of > 50 % (Huijnen et al., 2019). Urban measurement campaigns point to large discrepancies between modelled and observed OH abundances, for example in Lu et al., (Lu et al., 2013) who found a factor 2.6 difference in a campaign in the suburbs of Beijing.

The aim of this study is therefore to estimate the average OH concentration in the urban plume of large cities (hereafter referred

to as urban OH) from the downwind decay of the TROPOMI observed $NO_2/CO$ ratio. The proposed method makes use of the WRF model (Grell et al., 2005) to simulate the meteorological fields and atmospheric transport. The TROPOMI instrument (Veefkind et al., 2012), launched on 13 October 2017 on board the Sentinel-5 Precursor satellite, is particularly well suited for this task, as it measures both compounds with high sensitivity and spatial resolution. Our method uses CO, because it has a longer lifetime than $NO_2$ (weeks-months compared to a few hours). Therefore, CO can be considered as an inert tracer at the

time-scale of urban plumes. The difference in the rate of decay between $NO_2$ and CO provides therefore information about the photochemical oxidation of $NO_2$, because atmospheric dispersion is expected to have a very similar impact on both tracers and therefore cancels out in their ratio. The use of the $NO_2/CO$ ratio for estimating urban scale OH is further compared to the Exponentially Modified Gaussian function fit (EMG) method, using only satellite retrieved $NO_2$ (Beirle et al., 2011).

The city of Riyadh (24.63° N, 46.71° E ) is chosen as a test case. Riyadh is an isolated city and a strong source of CO and $NO_2$

pollution (Beirle et al., 2019; Lama et al., 2020). The frequent clear sky conditions over Riyadh yield a large number of valid TROPOMI CO and $NO_2$ data. The signal to noise in TROPOMI is high enough to detect the enhancement of CO and $NO_2$ over Riyadh in a single overpass (Lama et al., 2020). Model results from the Copernicus Atmospheric Monitoring Service (CAMS) for Riyadh show a distinct seasonality in OH (see Fig S1), which we attempt to evaluate using TROPOMI data for summer and winter.

This paper is organized as follows: Section 2 describes the TROPOMI $NO_2$ and CO data, the WRF model setup that was used, and the optimization method that is used for estimating OH. Optimization results and comparisons between TROPOMI and WRF are presented in section 3, followed by a summary and conclusion of the main finding in section 4. Additional figures and information about the optimization method are provided in the Supplement.

## 2. Data and Method

### 2.1 TROPOMI NO2 tropospheric column

We used the offline TROPOMI level 2 tropospheric column $NO_2$ [mole m$^{-2}$] data from retrieval versions 1.2.x for 2018 and 1.3.x for 2019 available at https://s5phub.copernicus.eu; http://www.tropomi.eu (last access: 21 September, 2020). $NO_2$ data of versions 1.2.x and 1.3.x have minor processing differences such as removal of negative cloud fraction, better flagging and uncertainty estimation. However, they use the same retrieval algorithm applied to level-1b version 1.0.0 spectra (Babic et al., 2019) recorded by the TROPOMI UV-Vis module in the 405-465nm spectral range. The TROPOMI $NO_2$ DOAS software, developed at KNMI, is used for the processing of $NO_2$ slant column densities (van Geffen et al., 2019). The improved $NO_2$ DOMINO algorithm of Boersma et al. (2018) has been used to translate slant columns into tropospheric column densities. In this algorithm, stratospheric contributions are subtracted from the slant column densities and the residual tropospheric slant column density is converted to tropospheric vertical column density using the air mass factor (AMF). The AMF depends on the surface albedo, terrain height, cloud height , cloud fraction and a priori $NO_2$ profiles from TM5-MP at $1°\times1°$ (Eskes et al., 2018; Lorente et al., 2017). The comparison of MAX-DOAS ground based measurements in European cities shows that TROPOMI underestimates of $NO_2$ columns by 7 % to 29.7 % (Lambert et al., 2019). To reduce the differences between satellite and model, we re-calculated the AMF by replacing the tropospheric AMF based on TM5 simulated vertical $NO_2$ columns, with the WRF-chem equivalent (Lamsal et al., 2010; Boersma et al., 2016; Visser et al., 2019; Huijnen et al., 2010), using the equation provided in the Appendix A. After the AMF recalculation, the $NO_2$ vertical profiles are consistent between satellite and model. Furthermore, the use of WRF-Chem has the advantage that it resolves $NO_2$ gradients between urban and downwind regions better than the coarser resolution TM5-MP model (Russell et al., 2011; McLinden et al., 2014; Kuhlmann et al., 2015). During summer, the AMF recalculation increases TROPOMI $NO_2$ by 5 % to 10 % and in winter by 25 % to 30 % in the urban plume over Riyadh, whereas background areas are less affected (see Fig S2 ). The S5P-PAL reprocessed $NO_2$ data available at https://data-portal.s5p-pal.com/products/no2.html differs by 7.5 % to 10 % in summer ( June to October, 2018) and 13.5 % to 16 % in winter (November, 2018 to March, 2019) compared to the AMF recalculated TROPOMI $NO_2$ data used in this study. These differences have been used to quantify the systematic uncertainty of the $NO_2$ data and its contribution to the uncertainty in the NOx emission and lifetime derived using our method (see Table S1, S2 and S3).

### 2.2 TROPOMI CO

For CO, the offline level 2 CO data product version 1.2.2 has been used, available at https://s5phub.copernicus.eu (last access: 20 September, 2020). The SICOR algorithm is applied to TROPOMI 2.3 μm spectra to retrieve CO total column density [molec cm$^{-2}$] (Landgraf et al., 2016). The retrieval method is based on a profile scaling approach, in which TROPOMI-observed spectra are fitted by scaling a reference vertical profile of CO using the Tikhonov regularization technique (Borsdorff et al., 2014). The reference CO profile is obtained from the TM5 transport model (Krol et al., 2005). The averaging kernel (A)

quantifies the sensitivity of the retrieved total CO column to variations in the true vertical profile ($\rho_{true}$), as follows (Borsdorff et al., 2018a):

$$C_{retrieval} = A.\rho_{true} + \epsilon_{CO} \qquad\qquad (1)$$

where, $C_{retrieval}$ is the retrieved column average CO mixing ratio, $\epsilon_{CO}$ is the retrieval error, statistically represented by the retrieval uncertainty that is provided for each CO retrieval.

The comparison of TROPOMI derived XCO to the 28 different TCCON ground based station suggest that difference between TCCON and TROPOMI is in the range of $9.1 \pm 3.3$ % (Shah et al., 2020). Such difference is used to estimate the uncertainty in the NOx emission and life time (see Table S1, S2, S3 and Text S6).

**2.3 Satellite Data Selection and Filtering Criteria**

As $NO_2$ and CO are retrieved from different channels of TROPOMI using different retrieval algorithms, the filtering criteria
and spatial resolutions of CO and $NO_2$ are different. The data filtering makes use of the quality assurance value (qa) and is provided with the CO and $NO_2$ retrievals, ranging from 0 (no data) to 1 (high quality data). We selected $NO_2$ retrievals with qa $\geq 0.75$ (clear sky condition) and CO retrievals with qa $\geq 0.7$ (clear sky or low level cloud) as in Lama et al., (2020). The SICOR algorithm was originally developed for SCIAMACHY to account for the presence of low elevation clouds, increasing the number of valid measurements (Borsdorff et al., 2018a). In addition, the CO stripe filtering technique is applied as described
by Borsdorff et al. (2018). Using dry air column density derived from the surface pressure data in CO and $NO_2$ TROPOMI files, the total CO column and tropospheric $NO_2$ column densities are converted to dry column mixing ratios XCO (ppb) and $XNO_2$ (ppb). The spatial resolution of the $NO_2$ data is finer compared to the CO data ($3.5x7$ $km^2$ versus $5.5x7$ $km^2$). After the CO and $NO_2$ retrievals pass the filtering criteria, their co-location is approximated by assigning the centre coordinates of an $NO_2$ retrieval to the CO footprint in which it is located (Lama et al., 2020).

**2.4 Weather Research Forecast model (WRF)**

We have used WRF- chemistry model (http://www.wrf-model.org/ ), version 3.9.1.1 to simulate $NO_2$ and CO mixing ratios over Riyadh. WRF is a non-hydrostatic model designed by the National Center for Environmental Protection (NCEP) for both atmospheric research and operational forecasting applications. For this study, we have setup three nested domains in the model at resolutions of 27 km, 9 km and 3 km, centred at 24.63°N, 46.71°E. The first and second domain cover Saudi Arabia and
provide the boundary conditions for the nested third domain (see Fig. S3). The analysis in this paper uses the 500 x 500 $km^2$ sub region around Riyadh in the third domain, containing 161 by 161 grid cells. All domains are extended vertically from the Earth's surface to 50 hPa, using 31 vertical layers, with 17 layers in the lowermost 1500 m. WRF simulations are performed using a time step of 90 seconds for the period June 2018 to March 2019, using a spin-up time of 10 days.

We have used the Unified Noah land surface model for surface physics (Ek et al., 2003; Tewari et al., 2004), an updated
version of the Yonsei University (YSU) boundary layer scheme (Hu et al., 2013) for the boundary layer processes, and the

Rapid Radiative Transfer Method (RRTM) for short-wave and long-wave radiation (Mlawer et al., 1997). Cloud physics is solved with the new Tiedtke cumulus parameterization scheme (Zhang and Wang, 2017). The WRF Single Moment 6-class scheme is used for microphysics (Hong and Lim, 2006). The WRF coupling with chemistry (WRF-chem) allows the simulation of tracer transport and the chemical transformation of trace gases and aerosols. Here, we used the passive tracer

transport function instead of the encoded chemistry in WRF to speed up the model simulation and reduce the computational cost. In addition, the passive tracer option helps in separating the influences of wind, OH and the rate constant of the $NO_2+OH$ reaction ($K_{NO2.OH}$) on the $NO_2/CO$ ratio in the downwind city plume. Compared to previously used methods (Beirle et al., 2011b; Valin et al., 2013) which did not use a transport model at all, we consider this an important improvement. The function of different tracers, their acronym and explanation of different WRF simulations is provided in Table 1.

The meteorological initial and boundary conditions are based on NCEP data at 1°x1° spatial and 6-hr temporal resolution available at https://rda.ucar.edu/datasets/ds083.2/. Nitrogen Oxides ($NO_x = NO_2 +NO$) and CO anthropogenic emissions have been taken from the Emission Database for Global Atmospheric Research v4.3.2 (EDGAR) 2012 at 0.1°x0.1° spatial resolution (Crippa et al., 2016). The EDGAR 2012 data have been re-gridded to the resolution of the WRF domains and hourly, weekly and monthly emission variations are taken into account using the temporal emission factors provided by van

der Gon et al. (2011). The chemical boundary conditions for CO and $NO_x$ are based on the CAMS chemical reanalysis product at 0.75°x0.75° spatial, and 3-hourly temporal resolution (Inness et al., 2019), retrieved from https://ads.atmosphere.copernicus.eu/cdsapp#!/dataset/cams-global-reanalysis-eac4?tab=form, last access: 1[st] November, 2020). XCO and $XNO_2$ boundary condition based on CAMS is assumed to be representative as background value within the domain. Since we do not explicitly compute the sources and sinks of background $NO_2$ inside the domain, we decide to

transport the boundary conditions as background passive tracers.

**Table 1. Summary of WRF simulations and the definition of tracers and acronym used.**

| WRF Simulation / Tracer | WRF input / Tracer definition |
|---|---|
| **Prior** | WRF run using NCEP meteorological data, EDGAR CO and $NO_x$ emissions, CAMS OH, and CAMS CO and NOx as initial and lateral boundary conditions. |
| **WRF$_{OH*1.1}$** | Prior run with CAMS OH increased by 10 % |
| **Optimized run$_{1st\ iter}$** | Optimized state (background, emission, OH) after iteration 1 |
| **Optimized run$_{2nd\ iter}$** | Optimized state (background, emission, OH) after iteration 2 |
| **CO** | |
| **XCO$_{emis}$** | The contribution of urban CO emissions to XCO |
| **XCO$_{Bg}$** | The contribution of the background to XCO |
| **XCO$_{WRF}$** | XCO from the Prior run |
| **XCO$_{WRF,1st\ iter}$** | XCO from Optimized run$_{1st\ iter}$ |

| | |
|---|---|
| **XCO $_{WRF,opt}$** | XCO from Optimized run$_{2nd\ iter}$ |
| **NO₂** | |
| **XNO$_{2\ emis}$** | The contribution of urban $NO_x$ emissions to $XNO_2$, ignoring the OH sink |
| **XNO$_{2\ (emis,OH)}$** | As $XNO_{2\ (emis)}$ accounting for the OH sink |
| **XNO$_{2\ (emis,OH*1.1)}$** | As $XNO_2(_{emis,OH})$ with CAMS OH increased by 10 % |
| **XNO$_{2\ Bg}$** | The contribution of the background to $XNO_2$ |
| **XNO$_{2\ WRF}$** | $XNO_2$ from the Prior run. |
| **XNO$_{2\ (WRF,OH*1.1)}$** | $XNO_2$ from $WRF_{OH*1.1}$. |
| **XNO$_{2\ WRF\ 1st\ iter}$** | $XNO_2$ from Optimized run$_{1st\ iter}$ |
| **XNO$_{2\ WRF\ opt}$** | $XNO_2$ from Optimized run$_{2nd\ iter}$ |
| **Ratio (NO₂/CO)** | |
| **Ratio$_{without\ OH}$** | Ratio of $XNO_{2\ emis}$ and $XCO_{emis}$ |
| **Ratio$_{with\ OH}$** | Ratio of $XNO_{2\ (emis,OH)}$ and $XCO_{emis}$ |
| **Ratio$_{Bg}$** | Ratio of $XNO_{2\ Bg}$ and $XCO_{Bg}$ |
| **WRF Ratio** | Ratio of $XNO_{2\ WRF}$ and $XCO_{WRF}$ |
| **WRF Ratio$_{OH*1.1}$** | Ratio of $XNO_{2\ (WRF,OH*1.1)}$ and $XCO_{WRF}$ |
| **WRF Ratio$_{1st\ iter}$** | Ratio of $XNO_{2\ WRF,1st\ iter}$ and $XCO_{WRF,1st\ iter}$ |
| **WRF Ratio$_{opt}$** | Ratio of $XNO_{2\ WRF,opt}$ and $XCO_{WRF,opt}$ |

The atmospheric transport in WRF causes the influence of $NO_x$ and CO emissions from Riyadh on their column average mixing ratios to be linear. Instead of a simplified photochemistry solver, we make use of a WRF-chem module for passive tracer transport for transporting NOx. This WRF module has been modified to account for the first order loss of NOx in reaction of $NO_2$ with OH, using NOx/$NO_2$ ratios from CAMS to translate NOx into $NO_2$ and CAMS OH fields to compute the chemical transformation of $NO_2$ to $HNO_3$ (see Text S1 for detail ).

In addition, we account for the chemical transformation of $NO_x$ to $HNO_3$ in the reaction of $NO_2$ with OH. This is a simplified treatment of the lifetime of $NO_x$ as other photochemical pathways play a role, such as:

- The oxidation of $NO_2$ in reaction with organic radicals ($RO_2$) to form the alkyl and multifunctional nitrates ($RONO_2$) (Romer Present et al., 2019)
- NOx loss due to the formation dinitrogen pentoxide ($N_2O_5$) followed by heterogeneous transformation to $HNO_3$ (Shah et al., 2020).
- Peroxyacetyl nitrate (PAN) formation in equilibrium between $NO_2$ and the peroxyacetyl radical (Moxim, 1996).
- The dry deposition of $NO_2$ on the surface and plant stomata (Delaria et al., 2020).

The loss of $NO_2$ by OH to $HNO_3$ accounts for 60 % of the global NOx emission (Stavrakou et al., 2013). Macintyre and Evans.,(2010) showed that the $N_2O_5$ pathway reduces NOx concentrations by 10 % in the tropics ($30^o$ N to $30^o$ S) and 40 % at northern latitudes. The NOx loss through $N_2O_5$ hydrolysis is largest at northern latitudes during winter (50 % to 150 %), unlike the tropics where its seasonality is small. Moreover, the removal of $N_2O_5$ is primarily important during night time because of its photolysis during daytime, whereas our analysis focuses on the midday overpass time (13:30) of TROPOMI when OH abundances are highest. For these reasons, we consider it safe to neglect the loss of $NO_x$ through $N_2O_5$ in our analysis for Riyadh. The dry deposition flux is also expected to be low as it is controlled largely by stomatal uptake, which is assumed to be insignificant for the low vegetation cover of Riyadh. The same is expected to be true for PAN formation because of its thermal decomposition at increasing temperatures. We acknowledge that our OH estimates should be regarded as upper limits due to the neglect of other NOx transformation pathways. A quantification of the combined effect would require full chemistry simulations, which we consider outside of the scope of this paper.

Note that in this study, OH is only applied to the urban NOx emission tracer ($XNO_{x\,emis}$). The CAMS $NO_x$ background tracer ($XNO_{x\,Bg}$) is transported in WRF without OH decay, since it already represents the balance between regional sources and sinks. CAMS hydroxyl radical (OH) data at a resolution of 0.75° x 0.75° spatial and 3 hourly temporal resolution (Inness et al., 2019) retrieved at https://ads.atmosphere.copernicus.eu/cdsapp#!/dataset/cams-global-reanalysis-eac4?tab=form, last access: 1st July, 2020) is spatially, temporally and vertically interpolated to the WRF grid. The $NO_x$ lifetime is derived as follows:

$$\frac{dNO_2}{dt} \ = \ K_{NO2\,OH}.[OH].[NO_2\ ] \qquad\qquad (2)$$

$$fact = \frac{NO_x}{NO_2} \qquad\qquad (3)$$

$$\tau_{NOx} = \frac{1}{\frac{K_{NO2\,OH}}{fact}.[OH]} \qquad\qquad (4)$$

where, $K_{NO2\,OH}$ is the International Union of Pure and Applied Chemistry (IUPAC) $2^{nd}$ order rate constant for the reaction of $NO_2$ with OH. "fact" represents the fractional contribution of $NO_2$ to $NO_x$ ($NO_x/NO_2$). This NOx to $NO_2$ conversion factor is derived from the CAMS reanalysis and re-gridded to WRF, to account for its spatial and temporal variation. $\tau_{NOx}$ is the lifetime of NOx.

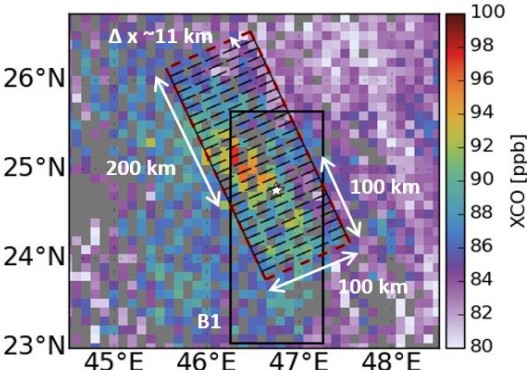 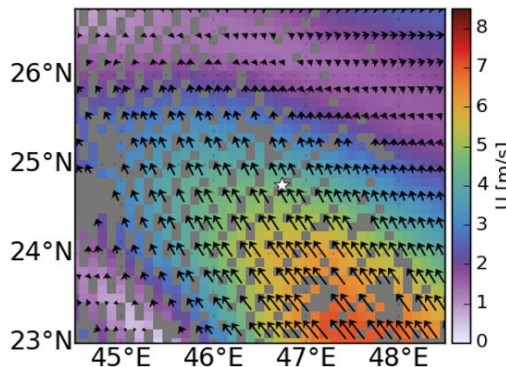

**Figure 1.** TROPOMI derived XCO (left) and average wind speed and wind direction from the surface to the top of boundary layer (right)  derived from the CAMS global reanalysis eac4 data at the TROPOMI overpass time over Riyadh for August 4$^{th}$, 2018. The white star represents the centre of Riyadh. The black box (B1) with a dimension of 300 x 100 km$^2$ is rotated in the average wind direction at 50 km radius from the centre of Riyadh at the TROPOMI overpass time resulting in the red box. For the calculation of cross-directional averaged NO$_2$ and CO, the red box is divided into 29 smaller cells with the width ($\Delta$x) ~11 km. For this TROPOMI derived XCO is gridded at 0.1°x0.1°.

The components of NOx (NO and NO$_2$) have short lifetimes during daytime because of the photo stationary equilibrium
exchanging NO and NO$_2$ into each other. For this reason, we estimate the lifetime of their sum (NO$_X$) which is determined largely by the reaction with OH. In earlier work with satellite NO$_2$ data, the Jet Propulsion Laboratory (JPL) high pressure limit was used as rate constant to represent the first order loss of NO$_2$ (Beirle et al., 2011; Lama et al., 2020; Lorente et al., 2019).  However, we found this approximation to be too crude, and therefore apply the full IUPAC recommended pressure dependent formula for the 2$^{nd}$ order rate constant.  Supplement Figure S4 shows the difference between the three rate constants,
i.e. JPL high pressure limit, JPL 2$^{nd}$ order and IUPAC 2$^{nd}$ order, confirming the importance of accounting for the pressure dependence.

WRF output for the third domain is interpolated spatially and temporally to the footprints of TROPOMI. The interpolated WRF- NOx tracers are converted to NO$_2$ using the conversion factor derived from the CAMS reanalysis accounting for its spatial and temporal variation (for the names and functions of tracers see Table 1). The averaging kernel available for each
TROPOMI CO and NO$_2$ observation is applied to the WRF output, after interpolation to the vertical layers of the TROPOMI retrieval. To compare WRF output to TROPOMI,  WRF derived XNO$_2$  (XNO$_{2\ WRF}$) is calculated by combining the NO$_2$ tracer that accounts for the OH effect (XNO$_{2\ (emis,OH)}$) and the CAMS NO$_2$ background  ( XNO$_{2\ Bg}$) (see Fig. S5 and S6) . Similarly, the CO emission tracer (XCO$_{emis}$) is added to the CAMS CO background (XCO$_{Bg}$) to calculate WRF simulated XCO (XCO$_{WRF}$ ) (see Fig. S7 and S8).

## 2.5 NO₂/CO ratio calculation using box rotation

The variation of the NO₂/CO ratio in the downwind city plume is calculate as a function of distance $x$ from the city centre in downwind direction. We select days with an average wind speed (U) in the range of 3.0 ms$^{-1}$ (Beirle et al., 2011) $< U < 8.5$ ms$^{-1}$ (Valin et al., 2013) within a 50 km radius from the centre of Riyadh (24.63° N, 46.71° E). The horizontal distribution of EDGAR emissions over Riyadh is used within this 50 km radius (Fig S9). Ninety five days in summer and 70 days in winter meet the wind speed criteria over Riyadh for the ratio calculation. The boundary layer average wind speed and direction is calculated using the CAMS global reanalysis eac4 (retrieved at https://ads.atmosphere.copernicus.eu/cdsapp#!/dataset/cams-global-reanalysis-eac4?tab=form , last access : 1$^{st}$ August, 2020) at a resolution of 0.75°x0.75° spatial and 3 hourly temporal resolution. For this, the CAMS wind vector is spatially and temporally interpolated to the central coordinate of TROPOMI pixels.

To compute the NO₂/CO ratio as function of the downwind distance $x$, TROPOMI and WRF data have been re-gridded at 0.1°x0.1°. A box (B1) is selected with a width of 100 km, from 100 km in upwind to 200 km in downwind direction of the city centre (see Fig 1a). The dimension of the box is motivated by multiple TROPOMI overpasses over Riyadh showing NO₂ and CO enhancements advected downwind over a ~200 km distance, without other large sources of NO₂ and CO within a 100 km radius of the city centre (see Fig. 1a). Figure 1(b) shows the boundary layer averaged wind speed and wind direction over Riyadh indicating flow towards the northeast on 4$^{th}$ of August, 2018. The box is rotated for every TROPOMI overpass depending upon the daily average wind direction within a 50 km radius from centre of Riyadh as shown in Figure 1(a) and Figure S10. The rotated box B1 is divided into N rectangular boxes, orthogonal to the wind direction with length ($\Delta$x) ~11 km (see Fig. 1 and Fig. S10). The XNO₂ and XCO grid cells that fall within the N rectangular boxes are selected to derive zonally averaged XNO₂ and XCO for summer and winter.

Unlike the enhancements over the city, $\Delta$XNO₂ and $\Delta$XCO become smaller than retrieval uncertainties at large distance from the city, where the ratio $\Delta$XNO₂/$\Delta$XCO becomes ill-defined. Therefore, we decided to use the ratio of mean XNO₂ and XCO instead of enhancements over the background. To analyse the influence of atmospheric transport and the OH sink on the WRF derived XNO₂/XCO ratio two different ratios are derived: 1. $\frac{XNO_{2\,emis}}{XCO_{emis}}$, named "Ratio$_{without\,OH}$", 2. $\frac{XNO_{2\,(emis,OH)}}{XCO_{emis}}$, named "Ratio$_{with\,OH}$"( see Table 1). The CAMS background accounts for the balance between regional source and sink in CTMs so it is excluded to analyse the influence of atmospheric transport on the ratio. For the comparison between TROPOMI and WRF, the CAMS backgrounds are included in "WRF RATIO" ($\frac{XNO_{2\,WRF}}{XCO_{WRF}}$) (see Table 1). The comparison of WRF RATIO to TROPOMI ratio, and the contribution of its components, is presented in section 3.2.

## 2.6 OH estimation: satellite data only

In the EMG method, following Beirle et al. (2011), 2D NO₂ column densities maps are assigned to eight equal wind sectors, spanning 360 ° for summer and winter. 1D column densities per wind sector are computed by averaging in cross wind direction.

This way, average $NO_2$ column density functions of the downwind distance to the city centre have been constructed for summer and winter (see Fig. S11). Using the EMG method as in Beirle et al., (2011), the e-folding distance x0 and $NO_2$ emissions have been estimated. The $NO_2$ lifetime is derived by dividing x0 by the average wind speed (5.46 ms$^{-1}$ and 5.24 ms$^{-1}$ for winter and summer, respectively) and is provided in Table 2. The OH concentration is derived from the inferred $NO_2$ lifetime using the IUPAC second order rate constant (for details see section Text S2 and S3). Rate constants at the time of TROPOMI overpasses are obtained from WRF by averaging the IUPAC second order rate constant from the surface to top of the planetary boundary layer. The PBL height at the time TROPOMI overpass has been taken from WRF. EMG derived $NO_2$ emissions are also converted to NOx emissions using the CAMS-derived conversion factor. Summer and winter averaged CAMS derived conversion factors for the box of 300 km x 100 km are 1.28 and 1.31, respectively.

## 2.7 OH estimation: WRF optimization

To jointly estimate the $NO_x$ and CO emissions as well as the OH concentration from the TROPOMI data, a least squares optimization method is used. This method fits the model to the data by minimizing a cost function (J) (see Text S4 for details). The reaction of $NO_2$ with OH introduces a non-linearity in the OH optimization. To account for this non-linearity, we linearize the problem around the a priori starting point, using small perturbations (10 %) $\Delta$background, $\Delta$emission, $\Delta$OH. The non-linear model is fitted to the observations, by optimizing scaling factors $f_{Bg}$, $f_{emis}$, $f_{OH}$ to the perturbation functions $\Delta$background, $\Delta$emission and $\Delta$OH, respectively. This process is repeated iteratively, updating the linearization point and re-computing the perturbation functions. The scaling factor $f_{emis}$, $f_{oh}$ and $f_{bg}$ represent the modification of the prior in percentage change.

We estimate OH by optimizing WRF with TROPOMI in two ways 1) optimizing the simulated $NO_2$/CO ratio using TROPOMI-derived ratios, named as "Ratio optimization" and 2) optimizing $NO_2$ and CO separately using TROPOMI derived XCO and $XNO_2$ named as "Component wise optimization". First the ratio optimization is described followed by the component wise optimization. Optimized ratios are derived as follows:

$$F_{TROPOMI} = F + \frac{\Delta F}{\Delta emis} * \frac{f_{emis}}{10} + \frac{\Delta F}{\Delta OH} * \frac{f_{OH}}{10} + \frac{\Delta F}{\Delta Bg} * \frac{f_{Bg}}{10} \quad (5)$$

$$F = \frac{XNO_{2\,WRF}}{XCO_{WRF}}$$

$$XNO_{2\,WRF} = XNO_{2\,(emis,OH)} + XNO_{2\,Bg} \quad (6)$$

$$XCO_{WRF} = XCO_{emis} + XCO_{Bg} \quad (7)$$

$$\frac{\Delta F}{\Delta emis} = \frac{XNO_{2\,(emis,OH)} * 1.05 + XNO_{2\,Bg}}{XCO_{emis} * 0.95 + XCO_{Bg}} - F \quad (8)$$

$$\frac{\Delta F}{\Delta OH} \qquad = \frac{XNO_{2\,(emis,OH*1.1)} + XNO_{2\,Bg}}{XCO_{emis} + XCO_{Bg}} - F \qquad\qquad (9)$$

$$\frac{\Delta F}{\Delta Bg} \qquad = \frac{XNO_{2\,(emis,OH)} + XNO_{2\,Bg} * 1.05}{XCO_{emis} + XCO_{Bg} * 0.95} - F \qquad\qquad (10)$$

Here, $F_{TROPOMI}$ is the TROPOMI derived $NO_2/CO$ ratio, F is the WRF Ratio, $\frac{\Delta F}{\Delta emis}$ is the change in F due to an increase in the

$NO_2$ emission by 5 % and a decrease in the CO emission by 5 %  $(1.05/0.95 = {\sim}10\,\%)$, $\frac{\Delta F}{\Delta OH}$ is the change in F due to an increase

in OH by 10 % and $\frac{\Delta F}{\Delta Bg}$ is the change in F due to an increase in the $XNO_2$ background by 5 % and a decrease in the CO

background by 5 %. $XNO_{2\,(emis,OH)}$ is the contribution of city NOx emissions to $XNO_2$ accounting for the OH sink, $XNO_{2\,Bg}$

is the $NO_2$ background. $XCO_{emis}$ is the contribution of the EDGAR city CO emissions to XCO and $XCO_{Bg}$ is the CO background

derived from CAMS. $XNO_{2\,WRF}$ and $XCO_{WRF}$ is the WRF derived $XNO_2$ and XCO respectively. $XNO_{2\,(emis,OH*\,1.1)}$ is the

contribution of city NOx emissions to $XNO_2$ after increasing CAMS OH by 10 %. The scaling factors $f_{emis}$, $f_{OH}$ and $f_{Bg}$ obtained

from the ratio optimization have been divided by 10 because $\frac{\Delta F}{\Delta emis}$, $\frac{\Delta F}{\Delta OH}$ and $\frac{\Delta F}{\Delta Bg}$ are defined as the change in F due to

modification of emission, OH and background by 10 %.

Although the ratio optimization is sensitive to the emission ratio and the OH sink of $NO_2$, it is not sensitive to the absolute

emissions of CO and $NO_2$. Therefore, we performed component-wise optimizations for XCO and $XNO_2$ to optimize absolute

emissions. We also compare the OH factor obtained from the ratio optimization and component-wise optimization to test the

robustness of the method. The optimized $XNO_2$ is derived using Eq. (11). XCO is optimized using the same equation but

without considering the OH sink (see Appendix B).

$$XNO_{2\,TROPOMI} = XNO_{2\,WRF} + \Delta XNO_{2\,emis} * \frac{f_{emis}}{10} + \Delta XNO_{2\,OH} * \frac{f_{OH}}{10} + \Delta XNO_{2\,Bg} * \frac{f_{Bg}}{10} \qquad (11)$$

$$\Delta XNO_{2\,emis} = XNO_{2\,(emis,OH)} * 1.10 - XNO_{2\,(emis,OH)} \qquad\qquad (12)$$

$$\Delta XNO_{2\,OH} = XNO_{2\,(emis,OH*\,1.1)} - XNO_{2\,(emis,OH)} \qquad\qquad (13)$$

$$\Delta XNO_{2\,Bg} = XNO_{2\,Bg} * 1.10 - XNO_{2\,Bg} \qquad\qquad (14)$$

Here, $XNO_{2\,TROPOMI}$ is the TROPOMI derived $XNO_2$, $XNO_{2\,WRF}$ is the WRF $XNO_2$. $\Delta XNO_{2\,emis}$ is the change in $XNO_2$ due

to an increase in emission by 10 %, $\Delta XNO_{2\,OH}$ is change in $XNO_2$ due to an increase in CAMS OH by 10 % and $\Delta XNO_{2\,Bg}$ is

a change in the background $XNO_2$ by 10 %. The scaling factors $f_{emis}$, $f_{OH}$ and $f_{Bg}$ are divided by a factor 10, because $\Delta XNO_{2\,emis}$,

$\Delta XNO_{2\,OH}$ and $\Delta XNO_{2\,Bg}$ are defined as 10 % changes in NOx emission, OH and background level.

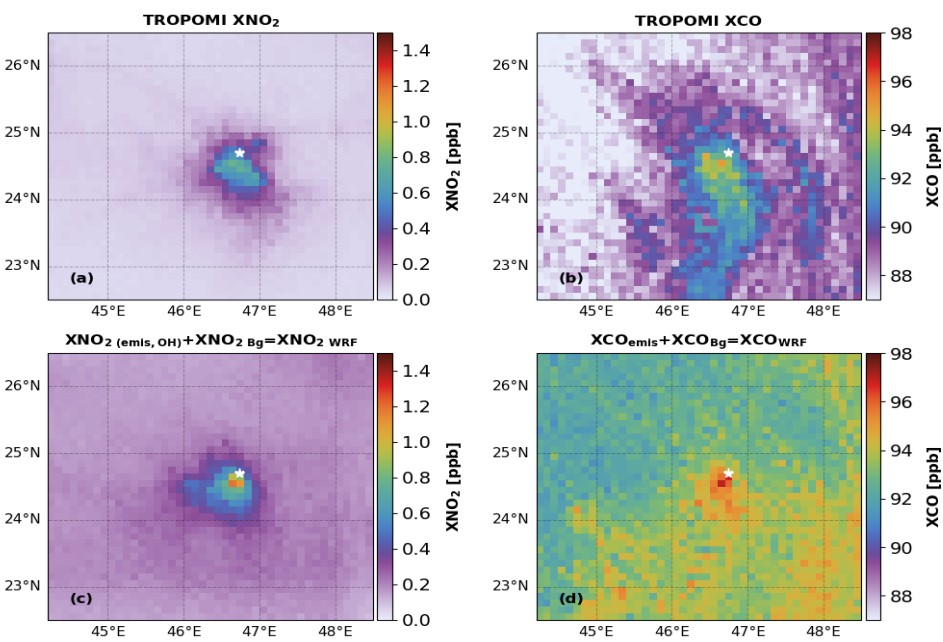

**Figure 2**. Comparison between XNO2 (left) and XCO (right) from TROPOMI and WRF over Riyadh averaged over June to October, 2018. Top panels show TROPOMI data and bottom panels the corresponding co-located WRF results. $XNO_{2\ WRF}$ is derived by adding $XNO_{2\ (emis,OH)}$ and $XNO_{2\ Bg}$ . $XCO_{WRF}$ is derived by adding $XCO_{emis}$ and $XCO_{Bg}$. The white star represents the centre of city. TROPOMI and WRF results are gridded at 0.1˚x0.1˚.

## 3. Results and Discussion

### 3.1. XNO₂ and XCO over Riyadh

In this subsection, we compare WRF-derived $XCO_{WRF}$ and $XNO_{2\ WRF}$ with TROPOMI for summer (see Fig. 2) and winter (see Fig. S6) over Riyadh. TROPOMI and WRF derived XCO and $XNO_2$ are averaged from June to October 2018 for summer and November 2018 to March 2019 for winter in a domain of 500 x 500 $km^2$ centered around Riyadh.

The comparison for summer in Figure 2 shows TROPOMI $NO_2$ after replacing the TM5-based tropospheric AMF with WRF profiles as described in Visser et al. (2019). The enhancement of $XNO_2$ and XCO over Riyadh due to urban emissions is clearly

separated from the background for TROPOMI and WRF, showing that the city of Riyadh is well suited to investigate the use of the $NO_2$/CO ratio to quantify OH in urban plumes. Due to the longer life-time of CO, the TROPOMI-observed XCO plume

extends further in the southeast direction compared to $XNO_2$. Figure 2 shows that our WRF simulations are able to reproduce the TROPOMI retrieved $XNO_2$ ($r^2 = 0.96$) and XCO ($r^2 = 0.78$) plumes, confirming that WRF-derived $\frac{XNO_{2\ WRF}}{XCO_{WRF}}$ is suitable for the optimization of CTM-derived OH concentrations using TROPOMI data. $XNO_{2\ WRF}$ is higher by 25 % compared to TROPOMI in the city centre. In the background, $XCO_{WRF}$ shows a similar spatial distribution as TROPOMI XCO, but the values are higher by 5 to 10 % (see Fig 2.). Close to the city centre, $XCO_{WRF}$ is ~5.7 % higher than TROPOMI XCO. In EDGAR 2011, emission sources are located in the centre of Riyadh (see Fig. S9). However, as noted by Beirle et al. (2019) they extend to a larger part of the city in reality. This difference in spatial distribution leads to higher $XNO_{2\ WRF}$ and $XCO_{WRF}$ close to centre of Riyadh compared to TROPOMI.

In winter, the wind direction is predominantly from the south easterly sector in WRF and TROPOMI (see Fig S12). The spatial distribution of $XCO_{WRF}$ ($r^2 = 0.73$) and $XNO_{2\ WRF}$ ($r^2 = 0.88$ ) matches quite well with TROPOMI. Therefore, the difference between summer and winter should offer the opportunity to quantify the seasonality in emissions and OH concentrations over Riyadh. In winter, $XCO_{WRF}$ is ~5 to 10 % higher than TROPOMI, while $XNO_{2\ WRF}$ is higher by 40 % to 50 %. The difference could either point to uncertainties in the $NO_2/CO$ emission ratio, uncertainties in the $NO_2$ lifetime, or inaccuracies in the background. By quantifying OH, we can evaluate these explanations (see section 3.3). $XNO_{2\ WRF}$ is higher by 20 % in winter than in summer. Contrary, TROPOMI $NO_2$ is lower by ~30 % in winter (Fig S12.) compared to summer (Fig. 2). Again, to disentangle the role of changing sources and sinks, we need an independent estimate of OH.

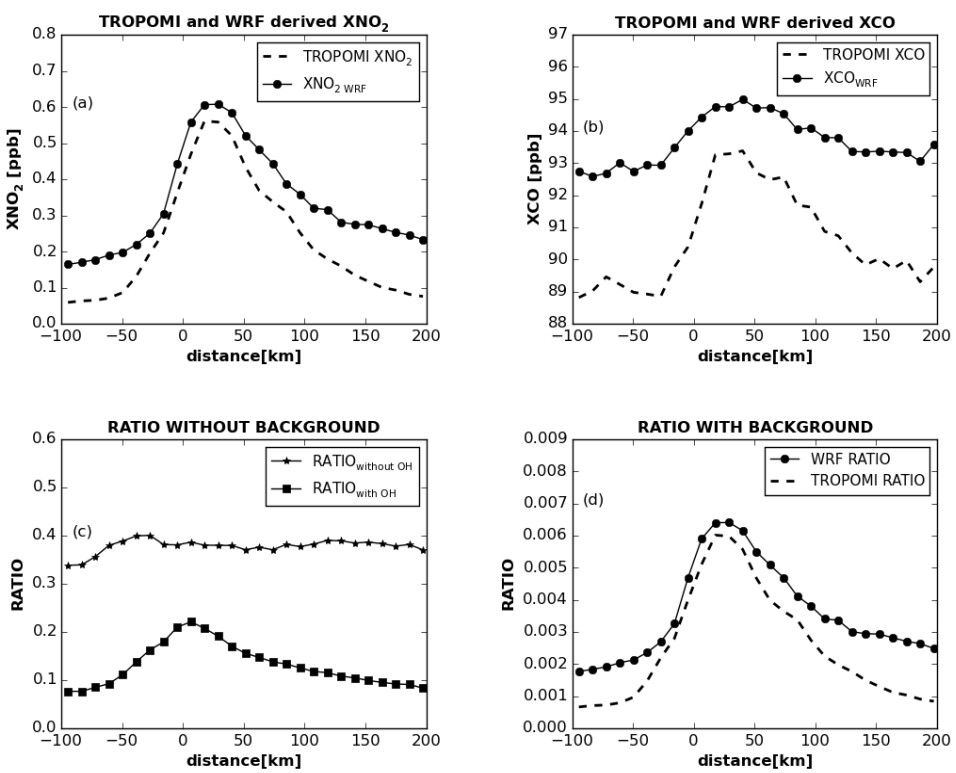

**Figure 3.** Comparison of WRF and TROPOMI averaged across the wind for each small box  a) $XNO_2$, b) XCO and c) WRF Ratio ($XNO_2$/ XCO) without CAMS background d) WRF Ratio ($XNO_2$/ XCO) with  background  and TROPOMI as a function of distance to the centre of Riyadh  for summer ( June, 2018  to October, 2018).

### 3.2. The $XNO_2$/XCO ratio and OH

Before comparing TROPOMI and WRF-derived $XNO_2$/XCO ratios, we first analyse the influence of atmospheric transport and the OH sink on the WRF derived $XNO_2$/XCO ratio. To do this three ratios are used 1. $Ratio_{without\ OH}$  2. $Ratio_{with\ OH}$  3. WRF RATIO  (see Table 1). As seen in Figure 3, S13 and S14,  WRF is able to reproduce the TROPOMI-observed downwind evolution of $XNO_2$ and XCO in summer and winter. The peak of the $XNO_2$ and XCO plumes is shifted away from the city centre due to the balance between the accumulation of urban emissions in the atmospheric column and atmospheric transport

(Lorente et al., 2019).

As expected, $Ratio_{without\ OH}$  shows an approximately straight line when the background is removed, because transport influences $NO_2$ and CO in the same way and therefore cancels out in the ratio (see Fig. 3b). The $Ratio_{with\ OH}$  however, shows an approximately Gaussian relation with distance due to the influence of the sink on $NO_2$. This comparison demonstrates the

sensitivity of the relation between $XNO_2/XCO$ ratio and downwind distance to the $NO_2$ lifetime, which we want to exploit to

quantify OH. When including the background, the shapes of the functions in Figure 3c change (not shown), because the relative weights of the background and city contributions to the ratio vary with distance of the city centre. In summer, the WRF RATIO is higher by ~15 % close to centre of city TROPOMI due to the overestimation of $XNO_{2\ WRF}$ in WRF (see Fig. 3d). However in the downwind plume, at a distance of 100 km WRF RATIO is higher by 20 % to 50 % compared to TROPOMI.

In winter, $Ratio_{without\ OH}$ and $Ratio_{with\ OH}$ show relations with downwind distance that are similar to summer, confirming

that an OH sink leads to a Gaussian structure of the ratio (see Fig. S14). The winter WRF RATIO is 40 % to 60 % higher than TROPOMI due to the overestimation of $XNO_2$ by 40 % to 50 %. The WRF RATIO close to the centre of city is also 20 % higher in winter than in summer, due to higher winter $XNO_{2\ WRF}$ than in summer (see Fig S12 and S15). In contrast, TROPOMI shows a higher ratio in summer compared to winter (see Fig S15). These differences between TROPOMI and WRF-derived ratios offer an opportunity to address uncertainties in CTM computed urban OH and emission inventories, which will be

explored next.

### 3.3 WRF optimization using synthetic data

To translate the discrepancies between TROPOMI and WRF derived ratios of section 3.2 into implied differences in emissions, OH and background, the least squares optimization method has been used as described in section 2.6. Before optimizing WRF using TROPOMI, pseudo data experiments in WRF have been carried out to test if the optimization method is capable of recovering true emissions and OH levels. To this end, changes in OH concentrations, emissions and background by known scaling factors have been applied to the WRF prior simulation to create a synthetic dataset. This process is repeated multiple times to create thousands of synthetic datasets. Subsequently, the scaling factors are obtained in the inversion procedure. These tests reveal that the estimation errors for $f_{emis}$, $f_{OH}$ and $f_{Bg}$ are less than 2.5 % (see Fig. S16). This confirms that the least square optimization method works, with two iterations leading to a sufficient accuracy, and can be used to estimate emissions and OH from TROPOMI data. Using TROPOMI data, estimation errors for $f_{emis}$, $f_{OH}$ and $f_{Bg}$ are expected to be higher due to atmospheric transport errors, simplified chemistry, and XCO and $XNO_2$ retrieval uncertainties . These errors did not play a role in the pseudo-data experiments, in which perfect transport and sampling was assumed.

To obtain a more realistic estimate of the uncertainty in least squares optimized OH, TROPOMI data have been replaced by $NO_2$, CO and $NO_2/CO$ ratio derived from WRF-chem using the Carbon Bond Mechanism Z (CBM-Z) gas-phase chemical mechanism (Zaveri and Peters, 1999). EDGAR based VOCs, NOx and CO emission have been used in combination with boundary condition for NO, $NO_2$, CO, ozone ($O_3$) from CAMS to run WRF-chem for August 17[th], 2018 and November 18[th], 2018 representing a summer and winter day, respectively. For August 17[th], 2018, the ratio and $XNO_2$ optimization increase the CAMS based prior OH of $1.19 \times 10^7$ molecules$cm^{-3}$ by 15.7 % and 13.4 %, respectively (see Fig S17). In the fully coupled online chemistry with WRF simulation, the boundary layer averaged OH for the box of 300 km x 100 km amounts to $1.33 \times 10^7$ molecules$cm^{-3}$, which is <5 % lower than the optimized OH value that is derived using our method. The optimized NOx and CO emissions differ by <11 % from the emission input used in the full chemistry version WRF. In winter, the optimization increases CAMS based OH of $1.03 \times 10^7$ molecules$cm^{-3}$ by 19.4 %. The OH derived from WRF with full online chemistry is $1.07 \times 10^7$ molecules$cm^{-3}$ and lower by 15.2 % than the optimized OH value. The component wise optimization increases the EDGAR NOx and CO emissions by 23.1 % and 10.5 %, respectively (see Fig S18). Overall, the uncertainty in optimized NOx, CO emission and OH derived from this test is <11 % in summer and 10 % to 23 % in winter. Since the lifetime of NOx is determined by other reactions in addition to the oxidation to $HNO_3$ considered in our method, it is expected to overestimate the real OH value. The test using WRF full chemistry confirms that this is indeed the case. The uncertainty for OH, NOx emission and CO emission are in good agreement with the CLASS computations explained in detail in Text S6.

## 3.4 WRF optimization using seasonally averaged TROPOMI data

The results for summer are summarized in Figure 4, showing the optimized fit to the TROPOMI data as well as the corresponding scaling factors $f_{emis}$, $f_{OH}$ and $f_{Bg}$ that are estimated. The optimized emission, OH and Bg obtained from the 2nd iteration is divided by prior to derive the $f_{emis}$, $f_{OH}$ and $f_{Bg}$ (see Text S5 for details). The convergence of the iterative procedure is shown in Fig S19 and S20 . The estimated uncertainties for the scaling factors $f_{emis}$, $f_{OH}$ and $f_{Bg}$ are derived by summing the contribution of wind speed, length and width of the box, $NO_2$ bias, CO bias and the different pathways of NOx loss in quadrature (see Text S6, Tables S1 and S2). For summer and winter, the uncertainties of the optimized OH concentrations is <17 % and < 29 % respectively. For NOx and CO emissions, the uncertainty is < 29 % in summer and winter. Figure 4a shows WRF ratios for summer in comparison to TROPOMI, before and after optimizing the OH concentration. The optimized WRF ratios fit the TROPOMI ratios well with $X^2 = 0.1$ (for the derivation of $X^2$ see section Text S7) .The prior and optimized

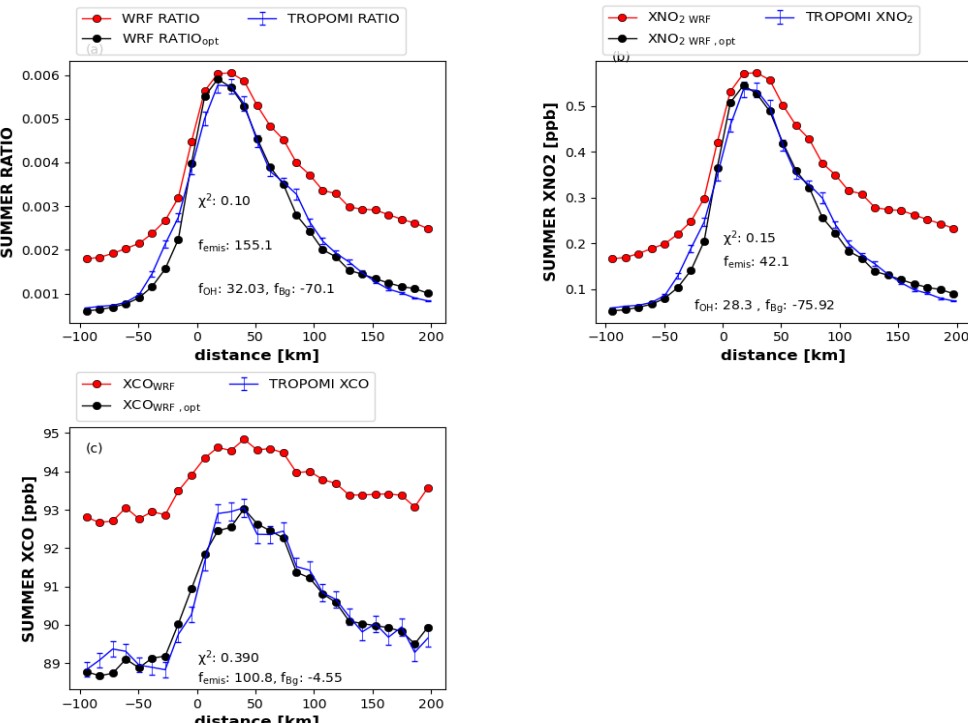

**Figure 4.** Comparison between TROPOMI and WRF, before and after optimization for Summer (averaged over June to October, 2018).  a) XNO2/XCO ratio, b) $XNO_2$ and c) XCO in comparison to TROPOMI. $f_{OH}$, $f_{emis}$ and $f_{Bg}$ are optimized scaling factors obtained iteratively for OH, emissions and background by least square optimization method. $f_{emis}$, $f_{OH}$ and $f_{Bg}$ are derived by accounting the total change in emission, OH and background using the corresponding scaling factors obtained from 1st and 2nd iterative step. The unit of scaling factor is in percent (%).

 emission ratio, OH concentration and background ratio obtained from component and ratio optimization for summer and winter is provided in Table S4. According to the ratio optimization, the emission ratio and CAMS OH are underestimated by $155 \pm 26$ % and $32 \pm 5.3$ % respectively (see Table S4). The optimized CAMS background ratio is lower by $70 \pm 6.5$ % compared to prior. It should be realized here that the ratio optimization does not estimate the absolute emission of $NO_2$ and CO, but only their ratio.


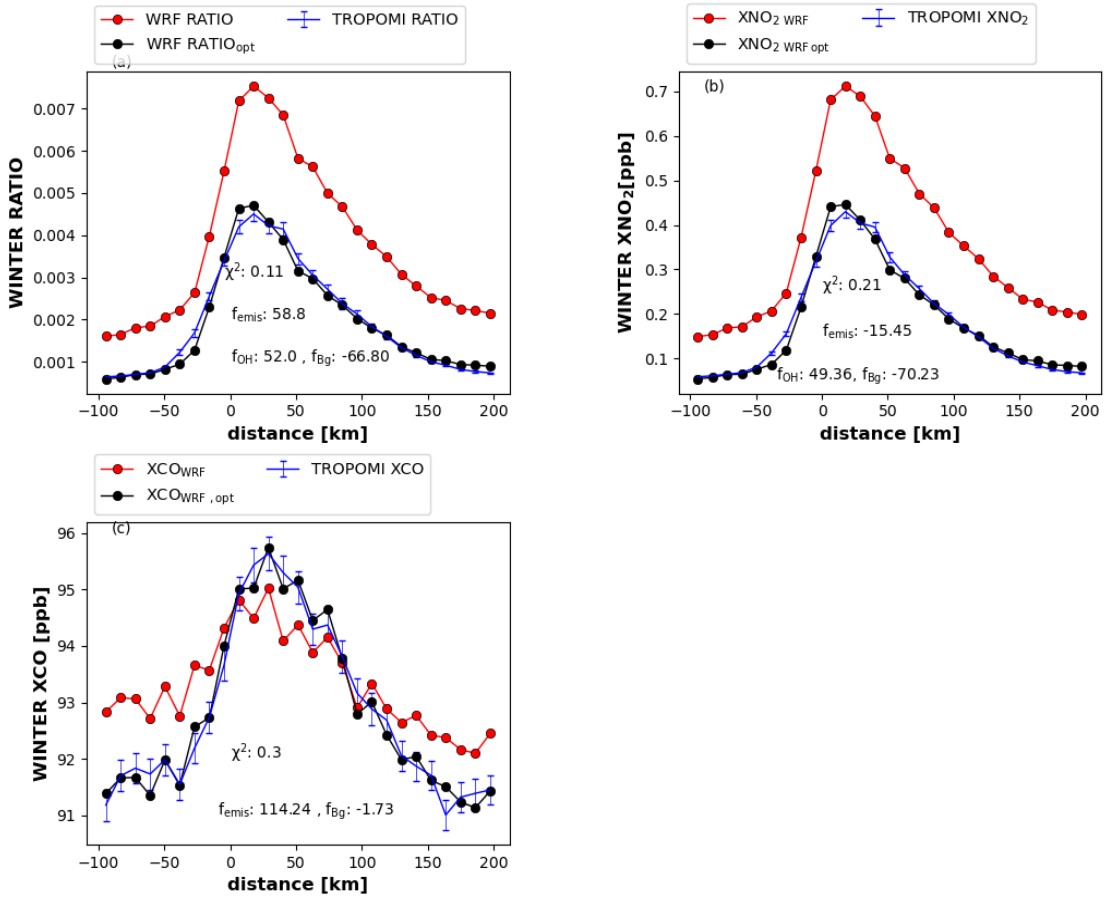

**Figure 5.** As Figure 4, for Winter (averaged over November, 2018 to March, 2019)

To derive the absolute emission, we performed component-wise optimizations of WRF-derived $XCO_{WRF}$ and $XNO_{2\ WRF}$. Optimized $XCO_{WRF}$ and $XNO_{2\ WRF}$ fit well to the TROPOMI data (see Fig. 4b and 4c). In the $XNO_2$ optimization, the EDGAR NOx emission is increased by $42.1 \pm 8.4$ % and the CAMs background is reduced by $75.9 \pm 10.0$ %. CAMS OH is increased

by 28.3 ± 3.9 % which is close to the results obtained from the ratio optimization (see Table S4). In the XCO optimization,

EDGAR CO emissions are roughly doubled and the background is reduced by 4.5 ± 0.7 % compared to CAMS (see Table S4).

The summer optimized NOx/CO emission ratio derived from the component wise optimization is 0.55 ± 0.09. The optimized emission ratio from ratio optimization is larger by factor 3.6  compared to component wise optimization (see Table S4). The difference between two estimates can be explained by different constraints on the solution in the two methods. In particular, the ratio inversion allows emission adjustment in a fixed relation between $NO_2$ and CO emissions whereas the component wise

has the full flexibility to adjust CO and $NO_2$ emission. The $NO_2$/CO ratio over a city is the sum of the contributions of the background and the city emission. The relative weight of the two is determined by the absolute background levels and absolute emissions of CO and $NO_2$.Therfore, the emission ratio estimated by ratio optimization is sensitive to the $XNO_{2Bg}$. However, the difference between the two estimates is larger than expected but does not affect the OH estimation. Lama et al., (2020) inferred an $NO_2$/CO emission ratio over Riyadh of 0.47 ± 0.1 for 2018 from TROPOMI favoring the Monitoring Atmospheric

Chemistry and Climate and CityZen (MACCity) emission ratio over that of EDGAR. The optimized emission ratio obtained from component wise optimization is consistent to Lama et al., (2020) and MACCity summer emissions. This shows that for the accurate estimation of the emission and emission ratio, the component wise optimization method is  preferable.

Figure 5 presents optimization results for winter, where optimized WRF is in similar good agreement with  TROPOMI as for summer with $X^2 = 0.11$ . For winter, the ratio optimization increases emission ratio by 58.8 ± 33 % and OH by 52.0 ± 14 %.

The ratio and component-wise optimizations again show similar OH adjustments, demonstrating the robustness of our method. The background ratio is reduced by 66.8 ± 11 %. The $XNO_2$ optimization reduces the EDGAR NOx emission by 15.45 ± 4.1 %  and the CAMS background by 70.2 ± 6.1 %. For XCO, the WRF $XCO_{Bg}$  is reduced by 1.74 ± 0.1 % in combination with a doubling of the EDGAR CO emission.  The optimized emission ratio (NOx/CO) derived from component wise optimization is 0.36 which is lower by 4.0  times than optimized emission ratio obtained from ratio optimization (see Table S4) .

**Table 2.** Overview of WRF optimized OH and NOx emissions for Riyadh and comparison to the EMG method. The estimated uncertainty for EMG and WRF derived NOx emission and OH concentration is the sum of the contribution of wind speed, length and width of box and $NO_2$ bias correction provided in Table S1, S2 and S3.

| Parameter | Summer WRF Optimization | | Summer EMG | Winter WRF Optimization | | Winter EMG |
|---|---|---|---|---|---|---|
| | Prior | Optimized | | Prior | Optimized | |
| **NOx emission (kg/second)** | 8.2 | 11.6±2.3 | 8.6±1.3 | 9.4 | 7.9±2.1 | 5.3±1.2 |
| **OH ($10^7$, molecules/cm³)** | 1.3 | 1.7 ± 0.32 | 1.53± 0.16 | 0.86 | 1.3 ±0.38 | 1.2± 0.16 |
| **NOx lifetime (hr)** | 3.1 | 2.4 ± 0.46 | 2.26 ± 0.3 | 4.9 | 3.3 ± 0.9 | 2.98 ± 0.4 |
| **NOx background (ppb)** | 0.22 | 0.053± 0.007 | 0.079±0.01 | 0.15 | 0.049±0.006 | 0.057±0.008 |

## 3.5 WRF optimization using a single TROPOMI overpass

To demonstrate the application of our WRF optimization method to single TROPOMI overpasses, results are presented in this subsection for August 18th, 2018. This date was selected for clear sky conditions with most of the TROPOMI $NO_2$ and CO pixels passing the data quality filter. During this day, the urban plume is transported in southwestern direction over Riyadh. The spatial distribution of $XNO_{2\,WRF}$ ($r^2 = 0.76$) and $XCO_{WRF}$ ($r^2 = 0.65$) matches quite well with TROPOMI (see Fig S21). The optimized ratio, $XNO_2$ and XCO for a single day fit well with TROPOMI ($X^2 = 0.1, 0.3$ and $0.7$) comparable to the summer averaged plumes indicating that the optimization method can be applied to single TROPOMI overpass. The ratio optimization increases the emission ratio and CAMS OH respectively by $111 \pm 18.4$ % and $37.9 \pm 6.2$ % respectively, whereas the background is reduced by $51.5 \pm 5.2$% (see Fig S22 a). The $XNO_2$ optimization increases the EDGAR NOx emission by $25.5 \pm 5.1$ % and CAMS OH by $32.3 \pm 4.4$ %, whereas the NOx background is reduced by $54.4 \pm 7.0$ % (see Fig S20 b). The CO optimization doubles the EDGAR CO emission and reduces the background by $6.1 \pm 0.97$ % (see Fig S20 c). The optimized NOx and CO emission for August 18th is $8.9 \pm 1.7$ kg/s and $18.9 \pm 4.0$ kg/s respectively and differs by <25 % with the summer optimized emission (see Table 2 and S5). The optimized OH derived from a single TROPOMI overpass is $1.73 \times 10^7 \pm 0.3$ molecules cm⁻³ differs by < 5 % from the summer averaged OH i.e. $1.7 \times 10^7 \pm 0.3$ moleculescm⁻³ confirming that the method yields realistic results for a single overpass.

**3.6 WRF optimization Vs  the EMG method**

To investigate the consistency between our method and the EMG method, the derived $NO_x$ lifetimes, emissions and OH concentrations using both methods are listed in Table 2 for winter and summer. Our optimization  and the EMG method agree well on the seasonal change in $NO_x$ emission and OH concentration. Both methods result in higher $NO_x$ emissions and shorter lifetimes in summer; lower NOx emissions and longer lifetimes in winter. Riyadh has dry and warm summer days and the increase in power consumption due to the use of air conditioning contributes to the higher emission in summer than in winter

(Lange et al., 2021). During the summer, EMG and the WRF optimization method both increase the NOx emission and OH concentration compared with the prior. The size of the $NO_x$ emission and OH concentration increase, obtained using the WRF optimization method is higher than the EMG method by 10%  to 29 %. However, the difference between the EMG method and the component optimization method are smaller compared to the uncertainty of the emission and OH concentration derived for the optimization method. For winter, the difference between the EMG and WRF-optimized results are smaller than the

difference between the EMG results and the prior . The NOx emission after optimization differs from the EMG method by 33 %. Optimized OH concentration and NOx lifetime differs by <10 % compared to EMG method. In general, the difference between the EMG and optimization results is within the uncertainty range of 20 to 30 %, confirming their consistency and strengthening the confidence in the estimates that are obtained from TROPOMI data. In contrast to EMG method, the optimization method can be used for a single TROPOMI overpass (see Section 3.6) and does not require yearly averaged $NO_2$

data, allowing analysis of day-by-day OH, NOx and CO emission (see Section 3.3). Segregation and averaging of $NO_2$ urban plume by wind sector is not required in the optimization method. The effect of transport cancels out in taking the $NO_2$ /CO ratio and loss of $NO_2$ is mostly governed by OH during the mid-day. In this study, NOx emission and OH concentration is estimated iteratively whereas the EMG method arrives at the solution in a single step.  However, since our optimization method requires a WRF model simulation it is computationally more expensive.  Uncertainties in transport may create mismatches

with the satellite observations, leading to errors in the optimized fit. This influences the quality of derived emission estimates (Dekker et al., 2017). Therefore, finding a simplified approach using satellite data to derive the emission ratio and to estimate OH concentration in urban plumes will be our focus in the future. In the future, the accuracy of our method can be further improved by accounting other NOx removal pathways.

**3.7 WRF optimized emissions and emission trends**

It should be realized that the a priori EDGAR emissions and TROPOMI optimized estimates represent different years (2012 and 2018, respectively). To check whether the emission differences that are found may be explained by trends in emissions, we compare EDGARv5.0  2012 NOx and CO emissions with 2018 accounting for seasonal and diurnal emission variations using temporal emission factors by van der Gon et al., (2011). EDGAR 2018 NOx and CO emissions are derived by linear extrapolation using emission from 2000 to 2015 (see Figure S23). For summer mid-day NOx emissions, the EDGAR emissions

increased by 16.7 % from 2012 to 2018, which is lower than our optimization results. For winter, mid-day NOx emissions

increase in EDGAR by 15.2 % from 2012 to 2018, whereas the WRF optimization yields reductions by 15.6%. In EDGAR, summer and winter CO emissions increased from 2012 to 2018 by 38.5 %. However, the WRF optimization suggests that the EDGAR CO emissions for summer and winter need to be doubled (see Table S4). Borsdorff et al., (2018b) mentioned that EDGAR CO emissions have to be increased significantly to match with TROPOMI CO observations over middle eastern cities

such as Tehran, Yerevan, Tabriz and Urmia. Overall , this points to a significant uncertainty in the EDGAR emission inventory at the city scale.

To test the accuracy of the linear extrapolation of EDGAR data, we compare the relative change in NOx and CO emission in 2012 to 2018 using CAMS Global (CAMS–GLOB) anthropogenic v4.2 emission datasets (https://ads.atmosphere.copernicus.eu/cdsapp#!/dataset/cams-global-emission-inventories?tab=overview). CAMS –GLOB

shows that for summer and winter NOx emission increases by 26 % from 2012 to 2018, which is higher by a factor 1.7 than EDGAR. CAMS-GLOB based summer and winter CO emission increases by 20 % from 2012 to 2018 which differs by ~40 % compared to EDGAR. In general, the relative increase in CO and NOx emission from EDGAR and CAMS-GLOB is much smaller compared to the difference with our optimization method.

## 4. Discussion

The TROPOMI retrieved $XNO_2/XCO$ ratio is useful for estimating mid-day OH over isolated localized sources, such as the city of Riyadh, showing a clear contrast between the urban plume and the background. Such TROPOMI derived OH estimates offer a new opportunity to evaluate urban photochemistry in chemistry transport models. OH depends non-linearly on NOx and VOC emission, meteorological conditions, etc.(Sillman, 1990) , which vary substantially between cities that are monitored by TROPOMI. Therefore, the application of our method to the global and multi-year dataset that is available could contribute

substantially to the understanding of urban photochemistry and the development of effective pollution mitigation strategies. In addition, the method requires local sources with $NO_2$ and CO emissions that are large enough to be detected by TROPOMI. Especially in European cities with lower CO emission where TROBabic, L., Braak, R., Dierssen, R., Kissi-Ameyaw, J., Kleipool, ´J., Leloux, J., Loots, E., Ludewig, A., Rozemeijer, N., Smeets, S., and Vacanti, G.: Algorithm theoretical basis document for the TROPOMI L01b data processor Erwin Loots Quintus Kleipool, 2017.

Beirle, S., Boersma, K. F., Platt, U., Lawrence, M. G., and Wagner, T.: Megacity emissions and lifetimes of nitrogen oxides probed from space, Science (80-. )., 333, 1737–1739, https://doi.org/10.1126/science.1207824, 2011a.

Beirle, S., Boersma, K. F., Platt, U., Lawrence, M. G., and Wagner, T.: Megacity emissions and lifetimes of nitrogen oxides probed from space, Science (80-. )., 333, 1737–1739, https://doi.org/10.1126/science.1207824, 2011b.

Beirle, S., Borger, C., Dörner, S., Li, A., Hu, Z., Liu, F., Wang, Y., and Wagner, T.: Pinpointing nitrogen oxide emissions

from space, Sci. Adv., 5, 1–7, https://doi.org/10.1126/sciadv.aax9800, 2019a.

Beirle, S., Borger, C., Dörner, S., Li, A., Hu, Z., Liu, F., Wang, Y., and Wagner, T.: Pinpointing nitrogen oxide emissions

from space, Sci. Adv., 5, https://doi.org/10.1126/sciadv.aax9800, 2019b.

Boersma, K. F., Vinken, G. C. M., and Eskes, H. J.: Representativeness errors in comparing chemistry transport and chemistry climate models with satellite UV-Vis tropospheric column retrievals, Geosci. Model Dev., 9, 875–898, https://doi.org/10.5194/gmd-9-875-2016, 2016.

Boersma, K. F., Eskes, H. J., Richter, A., De Smedt, I., Lorente, A., Beirle, S., Van Geffen, J. H. G. M., Zara, M., Peters, E., Van Roozendael, M., Wagner, T., Maasakkers, J. D., Van Der A, R. J., Nightingale, J., De Rudder, A., Irie, H., Pinardi, G., Lambert, J. C., and Compernolle, S. C.: Improving algorithms and uncertainty estimates for satellite NO2 retrievals: Results from the quality assurance for the essential climate variables (QA4ECV) project, Atmos. Meas. Tech., 11, 6651–6678, https://doi.org/10.5194/amt-11-6651-2018, 2018.

Borsdorff, T., Hasekamp, O. P., Wassmann, A., and Landgraf, J.: Insights into Tikhonov regularization: Application to trace gas column retrieval and the efficient calculation of total column averaging kernels, Atmos. Meas. Tech., 7, 523–535, https://doi.org/10.5194/amt-7-523-2014, 2014.

Borsdorff, T., Andrasec, J., De Brugh, J. A., Hu, H., Aben, I., and Landgraf, J.: Detection of carbon monoxide pollution from cities and wildfires on regional and urban scales: the benefit of CO column retrievals from SCIAMACHY 2.3 μm measurements under cloudy conditions, Atmos. Meas. Tech., 11, 2553–2565, https://doi.org/10.5194/amt-11-2553-2018, 2018a.

Borsdorff, T., Aan De Brugh, J., Hu, H., Hasekamp, O., Sussmann, R., Rettinger, M., Hase, F., Gross, J., Schneider, M., Garcia, O., Stremme, W., Grutter, M., Feist, Di. G., Arnold, S. G., De Mazière, M., Kumar Sha, M., Pollard, D. F., Kiel, M., Roehl, C., Wennberg, P. O., Toon, G. C., and Landgraf, J.: Mapping carbon monoxide pollution from space down to city scales with daily global coverage, Atmos. Meas. Tech., 11, 5507–5518, https://doi.org/10.5194/amt-11-5507-2018, 2018b.

Borsdorff, T., Aan de Brugh, J., Hu, H., Aben, I., Hasekamp, O., and Landgraf, J.: Measuring Carbon Monoxide With TROPOMI: First Results and a Comparison With ECMWF-IFS Analysis Data, Geophys. Res. Lett., 45, 2826–2832, https://doi.org/10.1002/2018GL077045, 2018c.

Borsdorff, T., Aan De Brugh, J., Pandey, S., Hasekamp, O., Aben, I., Houweling, S., and Landgraf, J.: Carbon monoxide air pollution on sub-city scales and along arterial roads detected by the Tropospheric Monitoring Instrument, Atmos. Chem. Phys., 19, 3579–3588, https://doi.org/10.5194/acp-19-3579-2019, 2019.

Browne, E. C., Min, K. E., Wooldridge, P. J., Apel, E., Blake, D. R., Brune, W. H., Cantrell, C. A., Cubison, M. J., Diskin, G. S., Jimenez, J. L., Weinheimer, A. J., Wennberg, P. O., Wisthaler, A., and Cohen, R. C.: Observations of total RONO2 over the boreal forest: NO x sinks and HNO3 sources, Atmos. Chem. Phys., https://doi.org/10.5194/acp-13-4543-2013, 2013.

Burnett, R., Chen, H., Szyszkowicz, M., Fann, N., Hubbell, B., Pope, C. A., Apte, J. S., Brauer, M., Cohen, A., Weichenthal, S., Coggins, J., Di, Q., Brunekreef, B., Frostad, J., Lim, S. S., Kan, H., Walker, K. D., Thurston, G. D., Hayes, R. B., Lim, C. C., Turner, M. C., Jerrett, M., Krewski, D., Gapstur, S. M., Diver, W. R., Ostro, B., Goldberg, D., Crouse, D. L., Martin, R. V., Peters, P., Pinault, L., Tjepkema, M., Van Donkelaar, A., Villeneuve, P. J., Miller, A. B., Yin, P., Zhou, M., Wang, L., Janssen, N. A. H., Marra, M., Atkinson, R. W., Tsang, H., Thach, T. Q., Cannon, J. B., Allen, R. T., Hart, J. E., Laden, F.,

Cesaroni, G., Forastiere, F., Weinmayr, G., Jaensch, A., Nagel, G., Concin, H., and Spadaro, J. V.: Global estimates of mortality associated with longterm exposure to outdoor fine particulate matter, Proc. Natl. Acad. Sci. U. S. A., 115, 9592–9597, https://doi.org/10.1073/pnas.1803222115, 2018.

Crippa, M., Janssens-Maenhout, G., Dentener, F., Guizzardi, D., Sindelarova, K., Muntean, M., Van Dingenen, R., and Granier, C.: Forty years of improvements in European air quality: Regional policy-industry interactions with global impacts, Atmos. Chem. Phys., https://doi.org/10.5194/acp-16-3825-2016, 2016.

Dekker, I. N., Houweling, S., Aben, I., Röckmann, T., Krol, M., Martínez-Alonso, S., Deeter, M. N., and Worden, H. M.: Quantification of CO emissions from the city of madrid using MOPITT satellite retrievals and WRF simulations, Atmos. Chem. Phys., 17, 14675–14694, https://doi.org/10.5194/acp-17-14675-2017, 2017.

Delaria, E. R., Place, B. K., Liu, A. X., and Cohen, R. C.: Laboratory measurements of stomatal NO2 deposition to native California trees and the role of forests in the NOx cycle, Atmos. Chem. Phys., 20, 14023–14041, https://doi.org/10.5194/acp-20-14023-2020, 2020.

Ding, J., Miyazaki, K., Van Der A, R. R., Mijling, B., Kurokawa, J. I., Cho, S. Y., Janssens-Maenhout, G., Zhang, Q., Liu, F., and Felicitas Levelt, P.: Intercomparison of NOx emission inventories over East Asia, Atmos. Chem. Phys., 17, 10125–10141,

https://doi.org/10.5194/acp-17-10125-2017, 2017.

Ek, M. B., Mitchell, K. E., Lin, Y., Rogers, E., Grunmann, P., Koren, V., Gayno, G., and Tarpley, J. D.: Implementation of Noah land surface model advances in the National Centers for Environmental Prediction operational mesoscale Eta model, J. Geophys. Res. Atmos., 108, 1–16, https://doi.org/10.1029/2002jd003296, 2003.

Eskes, H.J., van Geffen, J., Boersma, K.F., Eichmann, K.-U, Apituley, A., Pedergnana, M., Sneep, M., Pepijn, J., Loyola, D.:

Level 2 Product User Manual Henk Eskes, 2018.

Flagan, R. C. and Seinfeld, J. H.: Fundamentals of Air Pollution Engineering, 1988.

van Geffen, J. H. G. M., Eskes, H. J., Boersma, K. F., Maasakkers, J. D., and Veefkind, J. P.: TROPOMI ATBD of the total and tropospheric NO2 data products, S5P-KNMI-L2-0005-RP, issue 1.4.0, 6 Feburary 2019, 1–76, 2019.

Georgoulias, A. K., Van Der, R. A. J., Stammes, P., Folkert Boersma, K., and Eskes, H. J.: Trends and trend reversal detection

in 2 decades of tropospheric NO2 satellite observations, Atmos. Chem. Phys., 19, 6269–6294, https://doi.org/10.5194/acp-19-6269-2019, 2019.

van der Gon, H. D., Hendriks, C., Kuenen, J., Segers, A., and Visschedijk, A.: TNO Report: Description of current temporal emission patterns and sensitivity of predicted AQ for temporal emission patterns, 1–22, 2011.

de Gouw, J. A., Parrish, D. D., Brown, S. S., Edwards, P., Gilman, J. B., Graus, M., Hanisco, T. F., Kaiser, J., Keutsch, F. N.,

Kim, S.-W., Lerner, B. M., Neuman, J. A., Nowak, J. B., Pollack, I. B., Roberts, J. M., Ryerson, T. B., Veres, P. R., Warneke, C., and Wolfe, G. M.: Hydrocarbon Removal in Power Plant Plumes Shows Nitrogen Oxide Dependence of Hydroxyl Radicals, Geophys. Res. Lett., 0–2, https://doi.org/10.1029/2019GL083044, 2019.

Grell, G. A., Peckham, S. E., Schmitz, R., McKeen, S. A., Frost, G., Skamarock, W. C., and Eder, B.: Fully coupled "online" chemistry within the WRF model, Atmos. Environ., 39, 6957–6975, https://doi.org/10.1016/j.atmosenv.2005.04.027, 2005.

Guevara, M., Jorba, O., Tena, C., Denier van der Gon, H., Kuenen, J., Elguindi-Solmon, N., Darras, S., Granier, C., and Pérez García-Pando, C.: CAMS-TEMPO: global and European emission temporal profile maps for atmospheric chemistry modelling, Earth Syst. Sci. Data Discuss., 1, 1–60, 2020.

Hong, S. and Lim, J.: HongandLim_JKMS_WSM6_2006, http://www.mmm.ucar.edu/wrf/users/docs/WSM6-hong_and_lim_JKMS.pdf%5Cnhttp://search.koreanstudies.net/journal/thesis_name.asp?tname=kiss2002&key=2525908,

2006.

Hu, X. M., Klein, P. M., and Xue, M.: Evaluation of the updated YSU planetary boundary layer scheme within WRF for wind resource and air quality assessments, J. Geophys. Res. Atmos., 118, 10,490-10,505, https://doi.org/10.1002/jgrd.50823, 2013.

Huijnen, V., Eskes, H. J., Poupkou, A., Elbern, H., Boersma, K. F., Foret, G., Sofiev, M., Valdebenito, A., Flemming, J., Stein, O., Gross, A., Robertson, L., D'Isidoro, M., Kioutsioukis, I., Friese, E., Amstrup, B., Bergstrom, R., Strunk, A., Vira, J.,

Zyryanov, D., Maurizi, A., Melas, D., Peuch, V. H., and Zerefos, C.: Comparison of OMI NO2 tropospheric columns with an ensemble of global and European regional air quality models, Atmos. Chem. Phys., 10, 3273–3296, https://doi.org/10.5194/acp-10-3273-2010, 2010.

Huijnen, V., Pozzer, A., Arteta, J., Brasseur, G., Bouarar, I., Chabrillat, S., Christophe, Y., Doumbia, T., Flemming, J., Guth, J., Josse, B., Karydis, V. A., Marécal, V., and Pelletier, S.: Quantifying uncertainties due to chemistry modelling - Evaluation

of tropospheric composition simulations in the CAMS model (cycle 43R1), Geosci. Model Dev., 12, 1725–1752, https://doi.org/10.5194/gmd-12-1725-2019, 2019.

Ialongo, I., Virta, H., Eskes, H., Hovila, J., and Douros, J.: Comparison of TROPOMI/Sentinel-5 Precursor NO2 observations with ground-based measurements in Helsinki, Atmos. Meas. Tech., 13, 205–218, https://doi.org/10.5194/amt-13-205-2020, 2020.

Inness, A., Ades, M., Agustí-Panareda, A., Barr, J., Benedictow, A., Blechschmidt, A. M., Jose Dominguez, J., Engelen, R., Eskes, H., Flemming, J., Huijnen, V., Jones, L., Kipling, Z., Massart, S., Parrington, M., Peuch, V. H., Razinger, M., Remy, S., Schulz, M., and Suttie, M.: The CAMS reanalysis of atmospheric composition, Atmos. Chem. Phys., 19, 3515–3556, https://doi.org/10.5194/acp-19-3515-2019, 2019.

Krol, M., Houweling, S., Bregman, B., Broek, M. Van Den, Segers, A., Velthoven, P. Van, Peters, W., and Dentener, F.: and

Physics The two-way nested global chemistry-transport zoom model TM5 : algorithm and applications, Atmos. Chem. Phys., 417–432, 2005.

Kuhlmann, G., Lam, Y. F., Cheung, H. M., Hartl, A., Fung, J. C. H., Chan, P. W., and Wenig, M. O.: Development of a custom OMI NO2 data product for evaluating biases in a regional chemistry transport model, Atmos. Chem. Phys., 15, 5627–5644, https://doi.org/10.5194/acp-15-5627-2015, 2015.

Lama, S., Houweling, S., Folkert Boersma, K., Eskes, H., Aben, I., A. C. Denier Van Der Gon, H., C. Krol, M., Dolman, H., Borsdorff, T., and Lorente, A.: Quantifying burning efficiency in megacities using the NO2ĝ•CO ratio from the Tropospheric Monitoring Instrument (TROPOMI), Atmos. Chem. Phys., 20, 10295–10310, https://doi.org/10.5194/acp-20-10295-2020, 2020.

Lambert, J.-C., A. Keppens, D. Hubert, B. Langerock, K.-U. Eichmann, Q. Kleipool, M. Sneep, T. Verhoelst, T. Wagner, M. Weber, C. Ahn, A. Argyrouli, D. Balis, K.L. Chan, S. Compernolle, I. De Smedt, H. Eskes, A.M. Fjæraa, K. Garane, J.F. Gleason, F. Gouta, and P. W.: Sentinel-5 Precursor Mission Performance Centre Quarterly Validation Report of the Copernicus Sentinel-5 Precursor Operational Data Products # 03 : July 2018 – May 2019, 1–125, 2019.

Lamsal, L. N., Martin, R. V., Van Donkelaar, A., Celarier, E. A., Bucsela, E. J., Boersma, K. F., Dirksen, R., Luo, C., and Wang, Y.: Indirect validation of tropospheric nitrogen dioxide retrieved from the OMI satellite instrument: Insight into the seasonal variation of nitrogen oxides at northern midlatitudes, J. Geophys. Res. Atmos., 115, 1–15, https://doi.org/10.1029/2009JD013351, 2010.

Landgraf, J., Aan De Brugh, J., Scheepmaker, R., Borsdorff, T., Hu, H., Houweling, S., Butz, A., Aben, I., and Hasekamp, O.: Carbon monoxide total column retrievals from TROPOMI shortwave infrared measurements, Atmos. Meas. Tech., 9, 4955–4975, https://doi.org/10.5194/amt-9-4955-2016, 2016.

Lang, M. N., Gohm, A., and Wagner, J. S.: The impact of embedded valleys on daytime pollution transport over a mountain range, Atmos. Chem. Phys., https://doi.org/10.5194/acp-15-11981-2015, 2015.

Lange, K., Richter, A., and Burrows, J. P.: Variability of nitrogen oxide emission fluxes and lifetimes estimated from Sentinel-5P TROPOMI observations, Atmos. Chem. Phys. Discuss., 2, 1–32, https://doi.org/10.5194/acp-2021-273, 2021.

Liu, F., Beirle, S., Zhang, Q., Dörner, S., He, K., and Wagner, T.: NOx lifetimes and emissions of cities and power plants in polluted background estimated by satellite observations, Atmos. Chem. Phys., 16, 5283–5298, https://doi.org/10.5194/acp-16-5283-2016, 2016.

Lorente, A., Folkert Boersma, K., Yu, H., Dörner, S., Hilboll, A., Richter, A., Liu, M., Lamsal, L. N., Barkley, M., De Smedt, I., Van Roozendael, M., Wang, Y., Wagner, T., Beirle, S., Lin, J. T., Krotkov, N., Stammes, P., Wang, P., Eskes, H. J., and Krol, M.: Structural uncertainty in air mass factor calculation for NO2 and HCHO satellite retrievals, Atmos. Meas. Tech., 10, 759–782, https://doi.org/10.5194/amt-10-759-2017, 2017.

Lorente, A., Boersma, K. F., Eskes, H. J., Veefkind, J. P., van Geffen, J. H. G. M., de Zeeuw, M. B., Denier van der Gon, H. A. C., Beirle, S., and Krol, M. C.: Quantification of nitrogen oxides emissions from build-up of pollution over Paris with TROPOMI, Sci. Rep., 9, 1–10, https://doi.org/10.1038/s41598-019-56428-5, 2019a.

Lorente, A., Boersma, K. F., Eskes, H. J., Veefkind, J. P., van Geffen, J. H. G. M., de Zeeuw, M. B., Denier van der Gon, H. A. C., Beirle, S., and Krol, M. C.: Quantification of nitrogen oxides emissions from build-up of pollution over Paris with TROPOMI, Sci. Rep., https://doi.org/10.1038/s41598-019-56428-5, 2019b.

Lu, K. D., Hofzumahaus, A., Holland, F., Bohn, B., Brauers, T., Fuchs, H., Hu, M., Häseler, R., Kita, K., Kondo, Y., Li, X., Lou, S. R., Oebel, A., Shao, M., Zeng, L. M., Wahner, A., Zhu, T., Zhang, Y. H., and Rohrer, F.: Missing OH source in a suburban environment near Beijing: Observed and modelled OH and HO2 concentrations in summer 2006, Atmos. Chem. Phys., 13, 1057–1080, https://doi.org/10.5194/acp-13-1057-2013, 2013.

M. Tewari, F. Chen, W. Wang, J. Dudhia, M. A. L., K. Mitchell, M.Ek, G. G., and National, J. W. R. H. C.: Clay, Conf. Weather Anal. Forecast., n.d.

MacIntyre, H. L. and Evans, M. J.: Sensitivity of a global model to the uptake of N2O5 by tropospheric aerosol, Atmos. Chem. Phys., 10, 7409–7414, https://doi.org/10.5194/acp-10-7409-2010, 2010.

McLinden, C. A., Fioletov, V., Boersma, K. F., Kharol, S. K., Krotkov, N., Lamsal, L., Makar, P. A., Martin, R. V., Veefkind, J. P., and Yang, K.: Improved satellite retrievals of NO2 and SO2 over the Canadian oil sands and comparisons with surface measurements, Atmos. Chem. Phys., 14, 3637–3656, https://doi.org/10.5194/acp-14-3637-2014, 2014.

Mlawer, E. J., Taubman, S. J., Brown, P. D., Iacono, M. J., and Clough, S. A.: Radiative transfer for inhomogeneous atmospheres: RRTM, a validated correlated-k model for the longwave, J. Geophys. Res. Atmos., 102, 16663–16682,
https://doi.org/10.1029/97jd00237, 1997.

Monks, P. S., Granier, C., Fuzzi, S., Stohl, A., Williams, M. L., Akimoto, H., Amann, M., Baklanov, A., Baltensperger, U., Bey, I., Blake, N., Blake, R. S., Carslaw, K., Cooper, O. R., Dentener, F., Fowler, D., Fragkou, E., Frost, G. J., Generoso, S., Ginoux, P., Grewe, V., Guenther, A., Hansson, H. C., Henne, S., Hjorth, J., Hofzumahaus, A., Huntrieser, H., Isaksen, I. S. A., Jenkin, M. E., Kaiser, J., Kanakidou, M., Klimont, Z., Kulmala, M., Laj, P., Lawrence, M. G., Lee, J. D., Liousse, C.,
Maione, M., McFiggans, G., Metzger, A., Mieville, A., Moussiopoulos, N., Orlando, J. J., O'Dowd, C. D., Palmer, P. I., Parrish, D. D., Petzold, A., Platt, U., Pöschl, U., Prévôt, A. S. H., Reeves, C. E., Reimann, S., Rudich, Y., Sellegri, K., Steinbrecher, R., Simpson, D., ten Brink, H., Theloke, J., van der Werf, G. R., Vautard, R., Vestreng, V., Vlachokostas, C., and von Glasow, R.: Atmospheric composition change - global and regional air quality, Atmos. Environ., 43, 5268–5350, https://doi.org/10.1016/j.atmosenv.2009.08.021, 2009.

Moxim, W. J.: Simulated global tropospheric PAN: Its transport and impact on NOx, J. Geophys. Res. Atmos., 101, 12621–12638, https://doi.org/10.1029/96JD00338, 1996.

Pascal, M., Corso, M., Chanel, O., Declercq, C., Badaloni, C., Cesaroni, G., Henschel, S., Meister, K., Haluza, D., Martin-Olmedo, P., and Medina, S.: Assessing the public health impacts of urban air pollution in 25 European cities: Results of the Aphekom project, Sci. Total Environ., 449, 390–400, https://doi.org/10.1016/j.scitotenv.2013.01.077, 2013.

Pommier, M., McLinden, C. A., and Deeter, M.: Relative changes in CO emissions over megacities based on observations from space, Geophys. Res. Lett., 40, 3766–3771, https://doi.org/10.1002/grl.50704, 2013.

Romer, P. S., Duffey, K. C., Wooldridge, P. J., Edgerton, E., Baumann, K., Feiner, P. A., Miller, D. O., Brune, W. H., Koss, A. R., De Gouw, J. A., Misztal, P. K., Goldstein, A. H., and Cohen, R. C.: Effects of temperature-dependent NOx emissions on continental ozone production, Atmos. Chem. Phys., https://doi.org/10.5194/acp-18-2601-2018, 2018.

Romer Present, P. S., Zare, A., and Cohen, R. C.: The changing role of organic nitrates in the removal and transport of NOx, Atmos. Chem. Phys. Discuss., 1–18, https://doi.org/10.5194/acp-2019-471, 2019.

Russell, A. R., Perring, A. E., Valin, L. C., Bucsela, E. J., Browne, E. C., Wooldridge, P. J., and Cohen, R. C.: A high spatial resolution retrieval of NO2 column densities from OMI: Method and evaluation, Atmos. Chem. Phys., 11, 8543–8554, https://doi.org/10.5194/acp-11-8543-2011, 2011.

Sannigrahi, S., Kumar, P., Molter, A., Zhang, Q., Basu, B., Basu, A. S., and Pilla, F.: Examining the status of improved air quality in world cities due to COVID-19 led temporary reduction in anthropogenic emissions, Environ. Res., 196, 110927,

https://doi.org/10.1016/j.envres.2021.110927, 2021.

Shah, V., J. Jacob, D., Li, K., Silvern, R., Zhai, S., Liu, M., Lin, J., and Zhang, Q.: Effect of changing NO$x$ lifetime on the seasonality and long-term trends of satellite-observed tropospheric NO2 columns over China, Atmos. Chem. Phys., 20, 1483–1495, https://doi.org/10.5194/acp-20-1483-2020, 2020.

Sicard, P., Agathokleous, E., De Marco, A., Paoletti, E., and Calatayud, V.: Urban population exposure to air pollution in Europe over the last decades, Environ. Sci. Eur., 33, https://doi.org/10.1186/s12302-020-00450-2, 2021.

Sillman.S, Logan, A. and Wofsy, C.: The Sensitivity of Ozone to Nitrogen Oxides and Hydrocarbons in Regional Ozone Episodes 1 Now at of Atmospheric , Oceanic , and Space Sciences , al ., to explore the factors that influence ozone in Michigan , al areas during stagnation periods , again foc, Appl. Sci., 95, 1837–1851, 1990.

Sobanski, N., Thieser, J., Schuladen, J., Sauvage, C., Song, W., Williams, J., Lelieveld, J., and Crowley, J. N.: Day and night-time formation of organic nitrates at a forested mountain site in south-west Germany, Atmos. Chem. Phys., https://doi.org/10.5194/acp-17-4115-2017, 2017.

Stavrakou, T., Müller, J. F., Boersma, K. F., Van Der A., R. J., Kurokawa, J., Ohara, T., and Zhang, Q.: Key chemical NOx sink uncertainties and how they influence top-down emissions of nitrogen oxides, Atmos. Chem. Phys., 13, 9057–9082, https://doi.org/10.5194/acp-13-9057-2013, 2013.

United Nations: World Urbanization Prospects, 197–236 pp., https://doi.org/10.4054/demres.2005.12.9, 2018.

Valin, L. C., Russell, A. R., and Cohen, R. C.: Variations of OH radical in an urban plume inferred from NO 2 column measurements, Geophys. Res. Lett., https://doi.org/10.1002/grl.50267, 2013.

Veefkind, J. P., Aben, I., McMullan, K., Förster, H., de Vries, J., Otter, G., Claas, J., Eskes, H. J., de Haan, J. F., Kleipool, Q., van Weele, M., Hasekamp, O., Hoogeveen, R., Landgraf, J., Snel, R., Tol, P., Ingmann, P., Voors, R., Kruizinga, B., Vink, R., Visser, H., and Levelt, P. F.: TROPOMI on the ESA Sentinel-5 Precursor: A GMES mission for global observations of the atmospheric composition for climate, air quality and ozone layer applications, Remote Sens. Environ., https://doi.org/10.1016/j.rse.2011.09.027, 2012.

Verstraeten, W. W., Boersma, K. F., Douros, J., Williams, J. E., Eskes, H., Liu, F., Beirle, S., and Delcloo, A.: Top-down NOx emissions of european cities based on the downwind plume of modelled and space-borne tropospheric NO2 columns, 18, https://doi.org/10.3390/s18092893, 2018.

Visser, A. J., Folkert Boersma, K., Ganzeveld, L. N., and Krol, M. C.: European NOx emissions in WRF-Chem derived from OMI: Impacts on summertime surface ozone, Atmos. Chem. Phys., 19, 11821–11841, https://doi.org/10.5194/acp-19-11821-2019, 2019.

Wennberg, P. O., Stremme, W., Schneider, M., Feist, D. G., aan de Brugh, J., Toon, G. C., De Mazière, M., Gross, J., Grutter, M., Hu, H., Rettinger, M., Roehl, C., Pollard, D. F., Hasekamp, O., Hase, F., Borsdorff, T., Garcia, O., Kumar Sha, M., Arnold, S. G., Landgraf, J., Sussmann, R., and Kiel, M.: Mapping carbon monoxide pollution from space down to city scales with daily global coverage, Atmos. Meas. Tech., 11, 5507–5518, https://doi.org/10.5194/amt-11-5507-2018, 2018.

Zhang, C. and Wang, Y.: Projected future changes of tropical cyclone activity over the Western North and South Pacific in a

20-km-Mesh regional climate model, J. Clim., 30, 5923–5941, https://doi.org/10.1175/JCLI-D-16-0597.1, 2017.

POMI cannot detect the CO enhancement along with $NO_2$ this method cannot be applied.

We realise that our method only considers the first order loss of $NO_2$ by OH forming $HNO_3$. In reality, the $NO_2$ lifetime is influenced by more spatially and temporally varying factors such as temperature, ozone, and radiation (Lang et al., 2015; 725 Romer et al., 2018). In cities, the loss of $NO_2$ via the formation of alkyl and multifunctional nitrates ($RONO_2$) are also important reactions influencing the lifetime of $NO_2$ (Browne et al., 2013; Sobanski et al., 2017). For CO, secondary production from short-lived volatile organic compounds can also play an important role in urban pollution plumes. The application of full chemistry that includes all the sources and losses of $NO_2$ and CO could therefore further improve the accuracy of OH estimates. For cities at higher latitudes, especially in winter, it becomes more critical to account for the contribution of other pathways of 730 NOx loss than OH oxidation. Isolated tropical and subtropical cities are therefore best suited for application of our current method.

A sensitivity test has been performed in which $XNO_{x,Bg}$ is lost by OH. In this case the optimized NOx emission and OH for summer and winter differ by < 7 % from the default method where the background is treated as an inert tracer (see Table S6). Furthermore, a sensitivity test has been performed in which the prior emission has been changed. The optimized emission 735 varied by < 5 %, demonstrating robustness of the method to the choice of prior (see Fig S24). This also indicates that the optimization method can be used to study emission changes. Figure S25 shows that power plants and manufacturing industries are the largest pollutant emitter over Riyadh (Beirle et al., 2019).In this study, NOx and CO anthropogenic emissions are introduced at the surface, whereas the emission height of different sources is expected to vary in reality. The different emission heights for NOx and CO emission sources can also influence the result. In the future, realistic emission heights should also be 740 incorporated in WRF for accurate estimation of OH. Moreover, the temporal emission factors that have been used by van der Gon et al., (2011) are based on European countries. The comparison of van der Gon et al., (2011) with the Copernicus Atmosphere Monitoring Service TEMPOral profiles (CAMS-TEMPO) (Guevara et al., 2020) suggests that temporal emission factors for weekend road transport and monthly residential combustion are different in Riyadh compared to European countries CAMS-TEMPO is expected to provide a more accurate representation of emission variation due to the information on 745 temporal, spatial variations that is included. Road transport, CO emission has the largest contribution ~75 % to the total emission over Riyadh, whereas NOx emission from road contributes by 24 % to the total NOx emission. Residential combustion has the smallest contribution of ~0.3 to 0.4 % to total NOx and CO emissions (see Fig S25 ). In the future, the application of accurate diurnal emission factors for road transport (see Fig S26) can further improve the accuracy of urban OH concentrations estimated using TROPOMI derived $XNO_2/CO$ ratios. In addition, the seasonality for NOx and CO emissions 750 is different in Riyadh than in Europe, which should be accounted for in future studies also.

## 5. Conclusions

In this study, a new method is presented for estimating OH concentrations in urban plumes using TROPOMI observed $XNO_2/XCO$ ratios in combination with WRF simulations of the downwind pollution plume of large cities. Our new method has been tested for the city of Riyadh using synthetic as well as real TROPOMI data. Seasonal emissions and OH concentrations have been estimated for summer (June to October, 2018) and winter (Nov, 2018 to March, 2019).

WRF is well able to reproduce the spatial distribution of TROPOMI retrieved $XNO_2$ and XCO plumes over Riyadh during the summer and winter seasons. However, before the optimization, WRF overestimates $XNO_2$ by 15 % to 30 % in summer and 40 % to 50 % in winter compared to TROPOMI .. In both seasons, TROPOMI XCO agrees within 10 % with WRF. The WRF derived $XNO_2/XCO$ ratio is higher by 15 % to 30 % in summer and 40% to 60 % in winter compared to TROPOMI, explained mostly by differences in $XNO_2$.

The differences between WRF and TROPOMI observations have been used to optimize emissions and the $NO_2$ lifetime. To this end, scaling factors for the city emissions, OH and the background level have been optimized iteratively using a least squares method. Ratio and component wise optimizations have been compared to test the overall consistency of the method. In summer, the ratio and $XNO_2$ optimization for $XNO_2$ suggest that the OH prior from CAMS is underestimated by $32 \pm 5.3$ %. The OH estimates obtained from the ratio and $NO_2$-only optimization differs by <10 %, demonstrating the robustness of the method. Summertime emissions of NOx and CO from EDGAR are increased by $42.1\pm8.4\%$ and $101 \pm 21$ %. For winter, the ratio and component wise optimizations increase OH by $\sim52 \pm 14$ % to fit TROPOMI inferred ratios. In the optimization of winter data, NOx emissions are reduced by $15.5 \pm 4.1$ % and CO emissions are doubled. In the future, the remaining differences between TROPOMI observations and WRF simulations could be reduced further by the use of precise temporal and monthly emission factors, emission heights and full chemistry to account for secondary sources and sinks of CO and $NO_2$.

TROPOMI inferred OH concentrations obtained from the least squares optimization method have been compared to the EMG method. For the summer and winter, the optimized OH concentrations differ by 10% between two methods. These results confirm that urban emissions and OH concentrations can robustly be estimated from TROPOMI data. With our method, single TROPOMI overpasses can be used to estimate OH whereas EMG method requires averaging of urban $NO_2$ plume by wind sector. The iterative approach allows to test the factors i.e. $f_{emis}$, $f_{oh}$ and $f_{bg}$ obtained from optimization method, whereas EMG method does not allows such flexibility.

An important remaining uncertainty is the bias correction of the TROPOMI $XNO_2$ retrieval. Following the recommended procedure, the air mass factor AMF is recalculated by replacing the tropospheric AMF based on TM5, that is provided with the data, with WRF-chem. The TROPOMI $XNO_2$ bias correction increases the mixing ratio in the urban plume of Riyadh by 5 % to 10 % in summer and 25 % to 30 % in winter. The background is less affected by the bias correction. Without TROPOMI $XNO_2$ bias correction, the uncertainty in scaling factor for OH can vary up to 20 % and NOx emission to 60 % over Riyadh.

## Appendix A: AMF recalculation

The air mass factor (AMF) used in the retrieval of TROPOMI XNO$_2$ has been re-calculated by replacing the tropospheric AMF, calculated from the NO$_2$ column simulated by TM5, with its WRF-chem equivalent, as described by Lamsal et al. (2010) and Boersma et al. (2016) using the following Eq. (16),

$$M_{trop,\ WRF} = M_{trop,\ TM5} \times \frac{\sum_{l=1}^{L} A_{trop,l} x_{l,WRF}}{\sum_{l=1}^{L} x_{l,WRF}} \qquad (16)$$

where, $M_{trop,WRF}$ and $M_{trop,TM5}$ are the tropospheric air mass factors derived from WRF and TM5, respectively. $A_{trop,l}$ is the tropospheric averaging kernel, ranging from the surface to the uppermost layer of the troposphere in the TM5 model (l). $x_{l,WRF}$ is the equivalent NO$_2$ column density in model layer l, based on WRF. $A_{trop}$ in Eq. (16) is derived using $A_{trop} = A \times \frac{M}{M_{trop}}$, where M and $M_{trop}$ are the total and tropospheric AMF's respectively. Finally, the bias corrected NO$_2$ vertical column density is computed using,

$$NO_{2,\ bias\ corrected} = \frac{M_{trop,\ TM5}}{M_{trop,\ WRF}} \times NO_2$$

where, $NO_2$ is the TROPOMI tropospheric NO$_2$ vertical column density and $NO_{2,\ bias\ corrected}$ is the bias corrected TROPOMI tropospheric NO$_2$ vertical column density.

## Appendix B: XCO component wise optimization

The component wise optimization of XCO$_{WRF}$ to estimate the emission and background of CO uses the following equation,

$$XCO_{TROPOMI} = XCO_{WRF} + \Delta XCO_{emis} * \frac{f_{emis}}{10} + \Delta XCO_{Bg} * \frac{f_{Bg}}{10}$$

$$XCO_{WRF} = XCO_{emis} + XCO_{Bg}$$

$$\Delta XCO_{emis} = 0.10 * XCO_{emis}$$

$$\Delta XCO_{Bg} = 0.10 * XCO_{Bg}$$

Here, XCO$_{TROPOMI}$ is TROPOMI XCO, XCO$_{WRF}$ is the WRF simulated XCO accounting for emissions and background CO, XCO$_{emis}$ is the XCO contribution from the urban CO emission and XCO$_{Bg}$ is the CAMS-derived XCO background. $\Delta XCO_{emis}$ is the change in XCO due to emission and $\Delta XCO_{Bg}$ is the change in the XCO background level.

*Data Availability Statement.* TROPOMI CO and NO2 data can be downloaded from https://cophub.copernicus.eu/s5pexp.
EDGAR emission data is available at https://edgar.jrc.ec.europa.eu/emissions_data_and_maps. CAMS data can be
downloaded from https://ads.atmosphere.copernicus.eu/cdsapp#!/dataset/cams-global-reanalysis-eac4?tab=form. WRF
simulations output are available at https://zenodo.org/deposit?page=1&size=20

*Author contributions.* SL performed the data analysis, data interpretation, and wrote the paper. SH supervised the study. SH,
FKB, IA, MK and HACDG discussed the results. All co-authors commented on the paper and improved it.

*Competing interests.* The authors declare that they have no conflict of interest.

*Acknowledgments.*

We are thankful to the team that designed the TROPOMI instrument, consisting of the partnership between Airbus Defence
and Space Netherlands, KNMI, SRON, and TNO, commissioned by NSO and ESA. This work is supported by NWO GO
programme (grant no. 2017.036). We acknowledge the free availability of WRF-Chem model (http://www.wrf-model.org/ ).
Thanks to SURFSara for making the Cartesius HPC platform available for computations via computing grant no. 17235.

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
