# Peer review of "Estimation of OH in urban plumes using TROPOMI inferred NO2/CO"

_EGUsphere, 2022_

## Author Comment (AC1)

For presentation quality, I usually try to stay away from commenting on writing style, but I do think some re-organization would be beneficial to the reader. Instances that could help clarify confusion are noted under "Specific comments," but I might also suggest reframing the results in an "easier to digest" way. Currently, Section 3 follows the steps of the analysis quite closely, which gets quite overwhelming when discussing model vs. TROPOMI differences, then ratio optimization vs component wise differences and those differences vs. CAMS or EDGAR, then those differences vs. EMG, etc., each with an emissions, a background, and an OH component. Perhaps an easier to follow organization would first discuss emissions only, in terms of the evolution of the emissions over the course of the optimization, then OH, then background? This is only a suggestion, but I think it would improve the readability of the paper.

*Author Response:*

*Thank you for your time and suggestions, particularly concerning the structure of the paper. To clarify the structure, Section 3 has been divided into the following subsections*

*3.3 WRF optimization using synthetic data*

*3.4 WRF optimization using seasonally averaged TROPOMI data : Here, the ratio and component wise optimizations for summer are explained and compared. First we discuss emissions, followed by OH and background. Prior and optimized estimates are summarized in Table 2 to provide an easy overview of results.*

*3.5 WRF optimization using a single TROPOMI overpass*

*3.6 WRF optimization Vs the EMG method*

*3.7 WRF optimized emissions and emission trends*

*Table S4 has been added, summarizing the results of ratio and component wise optimizations for easy comparison.*

In terms of providing more context/motivation for this work, I would interested in seeing discussion on topics such as: how difficult would it be to apply this method to other cities? What are the limitations that might make this hard to do for some locations? How do these findings influence our understanding of urban pollution, or what role could they play in better quantifying emissions? Etc.

*Author Response :*

*The following sentence (line 482 to 485) has been added to the discussion section: " The TROPOMI retrieved $XNO_2/XCO$ ratio is useful for estimating mid-day OH over isolated localized sources, such as the city of Riyadh, showing a clear contrast between the urban plume and the background. In addition, the method requires local sources with $NO_2$ and CO emissions that are large enough to be detected by TROPOMI. Especially in European cities with lower CO emission where TROPOMI cannot detect the CO enhancement along with $NO_2$ this method cannot be applied. For cities at higher latitudes, especially in winter, it becomes more critical to account for the contribution of*

*other pathways of NOx loss than OH oxidation. Isolated tropical and subtropical cities are therefore best suited for application of our current method. "*

**Specific comments**

**L19: From the one-sentence description of the method in the abstract (that OH concentration, NOx and CO emissions are iteratively optimized), the referencing to "NO2/CO ratio optimization" and "XNO2 optimization" is unclear without having read the full paper. I would suggest clarifying further the concept of ratio and component-wise optimization. Also, aren't background conditions also optimized? This could be included in the method description.**

*Author Response:*

*The following sentence has been added (line 24 to 26): "The differences between model and TROPOMI are used to optimize the OH concentration, NOx ,CO emissions and their background iteratively using a least squares method. To estimate OH, WRF is optimized in two different ways using a) TROPOMI $NO_2/CO$ ratio to optimize the $XNO_2/XCO$ ratio in WRF b) TROPOMI derived $XNO_2$ and XCO to optimize WRF $XNO_2$ and XCO separately."*

*The optimization about the background conditions is provided in the method section.*

**L20: Again, on first reading of the abstract, the mention of CAMS comes as a surprise; I thought WRF was being used. Further e laboration on the method could help clarify.**

Author Response :

*The following has been added (line 15 to 18): "This study uses passive tracer transport in WRF-CHEM, , instead of the encoded chemistry in combination with auxiliary input variables such as Copernicus Atmospheric Monitoring Service (CAMS) OH , Emission Database for Global Atmospheric Research v4.3.2 (EDGAR) NOx , CO emissions and National Center for Environmental Protection (NCEP) based meteorological data. The CAMS based boundary condition for $NO_2$ and CO are considered as the representative background within the domain"*

**L30: Air pollution from cities doesn't just threaten the health of those living in the cities, but also populations downwind; this statement seems overly general.**

Author Response

*The sentence has been modified to "The associated rise in consumption of energy and materials leads to severe air pollution that is estimated to have caused premature death of 4 to 9 million people globally in 2015 (Sicard et al., 2021; Pascal et al., 2013; Burnett et al., 2018)."*

**L82: Please provide the months used in Fig. S1 (i.e., is summer the average of June-July-Aug?)**

**Author Response :**

*- Changed as suggested*

**L100: I believe the newer v.2.2.0 of the retrieval should help with the bias in $NO_2$ seen in the analysis, according to the statement here: http://www.tropomi.eu/data-products/nitrogen-dioxide. Is it feasible to try this analysis with the newer products? It is understandable that results**

cannot always be published immediately after they are produced, but if an update to the analysis cannot be undertaken, at least a discussion of how the analysis might be affected by newer data products or a suggestion for future directions should be included.

**Author Response :**

*The updated v.2.3.1 NO$_2$ data differ by 7.5 % to 10 % in summer and 13.5 to 16 % in winter compared to AMF re-calculated data. These differences are accounted for in the uncertainty calculation. This sentence has been added to Line 115: "The S5P-PAL reprocessed NO$_2$ data version 2.3.1 available at https://data-portal.s5p-pal.com/products/no2.html differs by 7.5% to 10 % in summer (June to October, 2018) and 13.5 % to 16 % in winter (November, 2018 to March, 2019) compared to the bias corrected TROPOMI NO$_2$ data used in this study. These differences have been used to quantify the systematic uncertainty of the NO$_2$ data and its contribution to the uncertainty in the NOx emission and lifetime derived using our method (see Table S1 and S2). "*

[Figure]

Figure R1. Comparison of TROPOMI derived XNO2 v2.3.1 Vs bias corrected XNO2 using AMF recalculation for summer (top) and winter (bottom) over Riyadh.

**L102: I'd be curious if the WRF-chem model does a better job of simulating urban NO2, in general, compared against TM5? So, is it fixing the bias issue for the right reasons?**

**Author Response:**

*WRF-Chem model resolves the gradients in $NO_2$ between urban and downwind regions, whereas TM5-MP smooths out such gradients at 1 deg x 1 deg resolution. Therefore, recalculating the AMF with WRF-Chem corrects for the underestimations in retrieved urban $NO_2$ columns and the overestimation over downwind regions which is incorporated in many studies (e.g. Huijnen et al., 2010; Visser et al., 2019; Douros et al., 2022).*

**L111: The authors assessed NO2 data quality vs ground-based measurements from prior studies; is there a similar analysis that can be done for CO? Or is there reason to believe that the reference CO profile from TM5 is more reliable than it was for NO2?**

**Author Response:**

*CO has the longer lifetime and background is much more important than for $NO_2$ which the low resolution model used for deriving prior can adequately resolve. Sha et al.,(2021) compared the TROPOMI derived XCO to the 28 different TCCON ground based station and concluded that average difference between TCCON and TROPOMI is in the range of 9.1 ± 3.3 % . Such difference is used to calculate the uncertainty (see Table  S1 and S2).*

**In Table 1, the term "XNO2(emis,OH)" is used in its own definition; I expecte it was intended to say "As XNO2(emis)…" – please check.**

Author Response:

*Changed as suggested*

**L181: I'm not sure I agree with this justification for not allowing XNOx,Bg to be lost by OH; NOx will continue to be oxidized, even if the plume it resides in was previously exposed to OH.  Is there any sort of sensitivity test that can be done to see how large an effect this would have on the results?**

**Author Response:**

*The comparison of background NOx in WRF and CAMS shows differences within 10 % to 20 % for summer and winter confirming that the chemical sources and sinks in background NOx are in approximate balance. The application of OH to the NOx background results in much smaller background NOx concentrations (50 % to 70 %) in WRF than in CAMS.*

*The sensitivity test performed for the summer and winter case in which the $XNO_{x,Bg}$ is lost by OH. The scaling factor for summer and winter NOx emission is lower by 18 % and 6% compared to the case using background without the OH effect. The optimized emission and OH for summer and winter obtained using $XNO_{x,Bg}$ with OH loss differs by less than 7 % compare to the case where background is treated as inert tracer (see Table R1).*

**Table R1. Overview of WRF optimized OH and NOx emissions for Riyadh using NOx background with and without the loss by OH.**

| Parameters | Summer | | | Winter | | |
|---|---|---|---|---|---|---|
| | Prior | Optimized using Bg without OH loss | Optimized using Bg with OH loss | Prior | Optimized using Bg without OH loss | Optimized using Bg with OH loss |

| | | | | | | |
|---|---|---|---|---|---|---|
| NOx emission (kg/s) | 8.2 | 11.6±2.4 | 11.1 | 9.4 | 7.9±1.8 | 8.03 |
| OH (1e7, molecules/cm3) | 1.3 | 1.7 ± 0.2 | 1.67 | 0.86 | 1.3 ±0.14 | 1.22 |

**L194: Please explain why the lifetime of NOx is the more relevant quantity to this analysis than the lifetime of NO2.**

**Author Response :**

*This sentence has been added at Line 211 " The components of NOx (NO and NO$_2$) have short lifetimes during daytime because of the photo stationary equilibrium exchanging NO and NO$_2$ into each other. For this reason, we estimate the lifetime of their sum (NO$_X$) which is determined largely by the reaction with OH. "*

**Figure 1 caption: Please indicate "(right)" to describe the right panel, presumably after "wind direction" or "boundary layer."**

**Author Response:**

- *Changed as suggested*

**L248: Is it possible that the NOx/NO2 conversion factor may not hold for emissions, since all NOx emissions from combustion processes occur in the form of NO, strictly speaking?   While NO converts relatively rapidly to NO2, this still might be something to consider.  Please discuss any anticipated implications of this assumption.**

**Author Response:**

*It's true that most of the NOx emission from combustion processes occurs in the form of NO. But in our case, we are looking at a much coarser resolution of 3 km x 3km, where the NO has largely been converted into NO$_2$ due to the fast photo-stationary equilibrium. Therefore, the NOx/NO$_2$ conversion factor, reflecting this equilibrium, is the best way to quantify the emitted amount of reactive nitrogen given the satellite observed amount of NO2. The conversion factor is derived from CAMS and varies temporally and spatially within the domain.*

**Fig. 4c: It seems very counterintuitive that the optimization for XCO increases emis by so much, barely decreases Bg, yet you still achieve a decline in the XCO quantities such that TROPOMI values are well matched.  Am I interpreting this correctly?**

**Author Response:**

*The CO background is much larger than the enhancement due to city emissions. Therefore, the small fractional reduction in the background has a larger impact on the XCO level than the emission increase.*

**Fig. 4 caption: How exactly are the f values shown here derived? It looks as though they are not simply the sum of f_1 and f_2 values shown in Fig. S17. Please either explain or point to the location in the text where this is explained.**

**Author Response:**

*This section has been added in the supplementary:*

*Text S5. Iterative scaling factor optimization*

*Step 1: Scaling factors $f_{OH1}$, $f_{emis1}$ and $f_{Bg1}$ (for OH, emissions and background levels) are derived from least squares optimization of WRF using a priori settings to TROPOMI.*

*Step 2: WRF is run with optimized inputs from Step 1 to derive $WRF\ Ratio_{1st\ iter}$, $XNO_{2\ 1st\ iter}$ and $XCO_{WRF,\ 1st\ iter}$.*

*Step 3: $f_{OH2}$, $f_{emis2}$ and $f_{Bg2}$ are derived as in Step 1 using the results from Step 2.*

*Step 4: Step 2 is repeated for $f_{OH2}$, $f_{emis2}$ and $f_{Bg2}$ to derive $WRF\ Ratio_{opt}$, $XNO_{2\ WRF,opt}$ and $XCO_{WRF,\ opt}$.*

*Step 5: Final optimized scaling factor are derived by multiplying the scaling factor from the $1^{st}$ and $2^{nd}$ iteration.*

**L345: I'm concerned that this test is more likely to work since you are dealing with an internally consistent system. Using the model, it is easier to be sure that it can replicate a hypothetical scenario posed in the model with enough adjustments. The real world and what TROPOMI are detecting could be very different systems, though, so if the model is missing underlying processes, there is less confidence that this optimization process is robust.**

**I suppose the pseudo data experiment is still worth doing, and I'm not sure what test I would suggest in its place, but perhaps some qualification should be added that the promising results of the experiment may stem from this being an ideal/consistent system.**

**Author Response:**

*To obtain a more realistic estimate of the uncertainty in least squares optimization derived OH, TROPOMI data have been replaced by $NO_2$, CO and $NO_2/CO$ ratio derived from WRF-chem using the Carbon Bond Mechanism Z (CBM-Z) gas-phase chemical mechanism (Zaveri and Peters, 1999). EDGAR based VOCs, NOx and CO emission have been used in combination with boundary condition for NO, $NO_2$, CO, ozone ($O_3$) from CAMS to run WRF-chem for August $17^{th}$, 2018 and November $18^{th}$, 2018 representing a summer and winter day, respectively.*

[Figure]

[Figure]

[Figure]

*Figure S17. WRF derived a) XNO₂/XCO, b) XNO₂ and c) XCO before and after optimization in comparison to WRF using full online chemistry with CBMZ chemical scheme for 17ᵗʰ August, 2018.*

*For August 17ᵗʰ, 2018, the ratio and XNO₂ optimization increase the CAMS based prior OH of $1.19 \times 10^7$ molecules/cm³ by 15.7 % and 13.4 %, respectively (see Figure S17). In the WRF-chem full online chemistry simulation the boundary layer averaged OH for the box of 300 km x 100 km amounts to $1.33 \times 10^7$ molecules/cm³, which <5 % lower than the optimized OH value that is derived using our method. The optimized NOx and CO emission differs by <11 % than the emission input in full online chemistry.*

*In winter, optimization increases CAMS based OH of $1.03 \times 10^7$ molecules/cm³ by 19.4%. The OH derived from WRF-chem full online chemistry is $1.07 \times 10^7$ molecules/cm³ and lower by 15.2% than the optimized OH value. The component wise optimization increases the EDGAR NOx and CO emissions by 23.1 % and 10.5 %, respectively. Overall, the uncertainty in optimized NOx, CO emission and OH derived from this test is <11 % in summer and 10 % to 23 % in winter. Since the lifetime of NOx is determined by other reactions in addition to the oxidation to HNO₃ considered in our method, it is expected to overestimate the real OH value. The test using WRF full chemistry confirms that this is indeed the case. The uncertainty for OH, NOx emission and CO emission are in good agreement with the CLASS computations explained in detail in Text S6.*

[Figure]

[Figure]

[Figure]

*Figure S18. Same as Figure S17 but for 18th November, 2018.*

**L352: I was initially confused that the f values in Figs. S17 and S18 changed so much between the first iteration and the second. I later realized that the second iteration values represented adjustments made to the first iteration values (i.e., f_emis doesn't go from being +158.5 to −1.3 from iteration 1 to 2 in Fig. S17a; it goes from 158.6 to 157.2, or however you derive the 155.1 f_emis in Fig. 4a). It may be worth describing this more fully, so other readers aren't confused.**

**Author Response:**

*As explained above already, section Text S5 has been added describing the iterative optimization method and how optimized scaling factors are derived from it.*

**Also, for Fig. S17c, please place the values of f_emis1 and f_Bg1 on the left side, 2nd iteration f's on the right, to avoid confusion. And, why is there not a green line in this panel corresponding to XCO_WRF,1st iter?**

**Author Response:**

*Changed as suggested.*

*In XCO optimization, the 1st iteration process does most of the job. The scaling factors from the second iteration are very close to the 1st iteration. Therefore, the green and black lines are on top of each other.*

**L371: It would be helpful to state the value from Lama et al. (2020) here.**

**Author Response:**

*-Changed as suggested*

**L417: Looking at Fig. S19, if this is done by linear extrapolation from data that is present for 2000-2015, why does year 2016 CO emissions drop followed by increases in 2017 and 2018?**

**Author Response:**

*Thank you for this comment, which pointed to a bug in the linear extrapolation which has been fixed. Here is the new figure:*

[Figure]

*Figure S19. EDGAR a) CO and b) NOx emission from 2000 to 2018 for summer and winter at the time TROPOMI overpasses over Riyadh. EDGAR 2000 to 2015 data is linearly extrapolated to derived emission data for 2018.*

**L426: Please state why this model simulation is well suited to evaluate emissions changes – how does it calculate emissions, if not by relying on the EDGAR inventory?**

**Author Response:**

*To test the dependency of the method on the EDGAR emission inventory, a sensitivity test is performed reducing the prior emission by factor 10 (see Figure R3). The optimized emission that we get from this simulation is equal to that of the base inversion. This shows that the optimized estimate is independent from the total emission from EDGAR. However, the optimized emission does rely on the spatial emission pattern, which is why we need a priori emissions.*

[Figure]

Figure R3. WRF derived $XCO_{emis}$ before and after reduction by factor 10 (left) is used to derive $XCO_{WRF}$ ($XCO_{emis}$ + $XCO_{Bg}$) and $XCO_{WRF, emis\ reduce\ by\ factor\ 10}$ ($XCO_{emis,reduce\ by\ factor\ 10}$ + $XCO_{Bg}$) (right). The comparison of TROPOMI derived XCO and $XCO_{WRF, emis\ reduce\ by\ factor\ 10}$ before and after optimization (right). The femis and fbg are the scaling factor for emission and background. The unit is in percentage.

**L447: What is CAMS-TEMPO based on? Is there a reason why its temporal emission factors for Riyadh should be especially trustworthy?**

**Author Response:**

*CAMS-TEMPO is based on statistical information on temporal emission variations per sector collected at the local, regional and national level. This information is used to derive meteorology-dependent parametrizations to understand the dependence on climatological and sociodemographic factors. CAMS-TEMPO is expected to provide a more accurate representation of emission variation due to the information on temporal, spatial variations that is included.*

**L464: Why give a range for summer but a precise value for winter?**

**Author Response:**

*The sentence is added " The WRF derived $XNO_2/XCO$ ratio is higher by 15 % to 30 % in summer and 40%  to 60 % in winter compared to TROPOMI, explained mostly by the difference in $XNO_2$."*

**-Changed as suggested**

**L470: "Estimates" here means estimates of OH change, correct?  Please clarify. (couldn't understand this part)**

**Author Response**

*The sentence is modified in Line 521 as follows "The OH estimates obtained from the ratio and $NO_2$-only optimization differ <10 %, demonstrating the robustness of the method."*

**L475: It's not just sources, but also some sinks are missing (for NO2), right? (this is done)**

**Author Response:**

- *Sinks has been added*

**L500: Is it possible to give a title to Appendix B, as was done for Appendix A? (this is done)**

**Author Response:**

- *The section is now called: XCO component wise optimization*

**L504-505: Why not write this in its simplified form, XCO_emis*0.10? The same goes for the next line. ( this is done)**

**Author Response:**

- *Changed as suggested*

**Technical corrections**

**L30: "threating" should be "threatening"**

- *changed as suggested*

**L65: Beginning "OH estimates from…" is not a complete sentence**

**L107: This URL returns a "Not found" message**

- *New URL https://s5phub.copernicus.eu has been added*

**L175: "save" should be "safe"**

- *Changed as suggested*

**L310: "emission" repeated twice**

- *Changed as suggested*

**L313: either "estimates" should be singular or "an" should be removed**

- *Changed as suggested*

**L355: f_B should be f_Bg**

- *Changed as suggested*

**L392: "a" should be removed, or else "days" should not be plural**

- *Changed as suggested*

**L397: "compare" should be "compared"**

- *Changed as suggested*

**L407: "it the solution" should be "at the solution"**

- *Changed as suggested*

**L419: "yield" should be "yields"**

 - *Changed as suggested*

**L422: "has" should be "have"**

- *Changed as suggested*

**L481: "allows" should be "allow"**

- *Changed as suggested*

**L509: Again, the link to the TROPOMI data appears to be invalid.**

- *Two links are provided for TROPOMI data which appear to work fine.*

**The Zenodo link for the WRF simulations requires a login, so I could not access the data; I'm unsure if this is typical or not.**

- *Zenodo doesn't require a login. The link directs to a folder with WRF files, including a README file with instructions for use.*

---

## Author Comment (AC2)

1. On line 162, the authors mention that they "account for the chemical transformation of NOx to HNO3 in the reaction of NO2 with OH." However, it isn't clear how this is done - whether a highly simplified mechanism was added to WRF-Chem or whether this was done offline. A more detailed explanation of this would be welcome, even if just in the supplement.

Author Response:

*The sentence is added in Line 187 "Instead of a simplified photochemistry solver, we make use of a WRF-Chem module for passive tracer transport for transporting NOx. This WRF module has been modified to account for the first order loss of NOx in reaction of $NO_2$ with OH, using $NOx/NO_2$ ratios from CAMS to translate NOx into $NO_2$ and CAMS OH fields to compute the chemical transformation of $NO_2$ to $HNO_3$ (see Text S1 for detail method).*

The larger issue is the choice to use passive tracers with this simplified chemistry rather than one of the established chemical mechanisms in WRF-Chem. Line 180 states that such simulations are considered outside the scope of this paper, but does not explain this reasoning. I could see two reasons for such a choice:

1. To reduce computational cost, making this easier to apply at scale. If this is the case, some measurements of the relative speedup compared to a full chemistry simulation would help support this choice.

2. The framework used in this paper required a specified OH background to permit the calculation of d[NOx]/d[OH] in a straightforward manner. With a full chemistry simulation, I suspect it would be much more difficult to impose a constant increase or decrease in OH for this purpose.

Whatever the reason for the choice to use passive tracers, I urge the authors to explain their reasoning behind this choice, given (as they mention) the potential impact of other NOx loss pathways.

Author Response:

*So the main motivation is indeed computing time, which for a single day is not restrictive, but to apply the method to longer-term averages or a list of TROPOMI observed cities it is. The sentence is added in Line 155 - 159 :*

*"Here, we used the passive tracer transport function instead of the encoded chemistry in WRF to speed up the model simulation and reduce the computational cost. In addition, the passive tracer option helps in separating the influences of meteorology, OH and the rate constant of the $NO_2+OH$ reaction ($K_{NO2.OH}$) on the $NO_2/CO$ ratio in the downwind city plume. Compared to previously used methods by Beirle et al., (2011);Valin et al.,(2011) which did not use a transport model at all, we consider this an important improvement. "*

Second, at line 404, the authors state that "the optimization method can be used for a single TROPOMI overpass and does not require yearly averaged NO2 data." This is contrasted with the EMG approach, which does need a significant amount of data to generate reliable results. However, the ability of the optimization method described in this paper to estimate OH for individual days is not clearly demonstrated in this paper. Since this seems to be one of the main advantages of the authors' optimization method over the EMG method, this should be shown in more detail. At least a timeseries plot of daily OH concentrations obtained by this method would help by showing that we

do get reasonable OH values with a single day of data. Further, I expect that there is a minimum amount of clear sky pixels over a city required for this method to work effectively. Assuming that clouds are uncommon over Riyadh, this could still be explored by withholding increasing percentages of the available pixels for a given day and testing how the estimated OH deviates with the reduction in data.

Author Response:

*We agree with the reviewer that it would have been better to demonstrate the application of our method to single satellite overpasses. The following subsection was added:*

**3.5 WRF optimization using a single TROPOMI overpass**

*To demonstrate the application of our WRF optimization method to single TROPOMI overpasses, results are presented in this subsection for August 18th, 2018. This date was selected for clear sky conditions with most of the TROPOMI $NO_2$ and CO pixels passing the data quality filter. During this day, the urban plume is transported in southwestern direction over Riyadh. The spatial distribution of $XNO_{2\ WRF}$ ($r^2 = 0.76$) and $XCO_{WRF}$ ($r^2 = 0.65$) matches quite well with TROPOMI (see Fig S21). The optimized ratio, $XNO_2$ and XCO for a single day fit well with TROPOMI ($X^2 = 0.1$, 0.3 and 0.7) comparable to the summer averaged plumes indicating that the optimization method can be applied to single TROPOMI overpass. The ratio optimization increases the emission ratio and CAMS OH respectively by 111 ± 18.4 % and 37.9 ± 6.2 % respectively, whereas the background is reduced by 51.5 ± 5.2% (see Fig S22 a). The $XNO_2$ optimization increases the EDGAR NOx emission by 25.5 ± 5.1 % and CAMS OH by 32.3 ± 4.4 %, whereas the NOx background is reduced by 54.4 ± 7.0 % (see Fig S20 b). The CO optimization doubles the EDGAR CO emission and reduces the background by 6.1 ± 0.97 % (see Fig S20 c). The optimized NOx and CO emission for August 18th is 8.9 ± 1.7 kg/s and 18.9 ± 4.0 kg/s respectively and differs by <25 % with the summer optimized emission (see Table 2 and S5). The optimized OH derived from a single TROPOMI overpass is $1.73 \times 10^7 \pm 0.3$ molecules cm$^{-3}$ differs by < 5 % from the summer averaged OH i.e. $1.7 \times 10^7 \pm 0.3$ moleculescm$^{-3}$ confirming that the method yields realistic results for a single overpass.*

[Figure]

**Figure S21.** Comparison between XNO2 (left) and XCO (right) from TROPOMI and WRF over Riyadh for 18[th] August , 2018 . Top panels show TROPOMI data and bottom panels the corresponding co-located WRF results. $\mathbf{XNO_{2\ WRF}}$ is derived by adding $\mathbf{XNO_{2\ (emis,OH)}}$ and $\mathbf{XNO_{2\ Bg}}$ . $\mathbf{XCO_{\ WRF}}$ is derived by adding $\mathbf{XCO_{emis}}$ and $\mathbf{XCO_{Bg}}$. The white star represents the centre of city. TROPOMI and WRF results are gridded at 0.1˚x0.1˚.

[Figure]

**Figure S22.** Comparison between TROPOMI and WRF, before and after optimization for 18th August, 2018. a) XNO2/XCO ratio, b) $XNO_2$ and c) XCO in comparison to TROPOMI. $f_{OH}$, $f_{emis}$ and $f_{Bg}$ are optimized scaling factors obtained iteratively for OH, emissions and background by least square optimization method. $f_{emis}$, $f_{OH}$ and $f_{Bg}$ are derived by accounting the total change in emission, OH and background using the corresponding scaling factors obtained from 1st and 2nd iterative step. The unit of scaling factor is in percent (%).

Third, the efforts to test this optimization approach described in the manuscript are a good foundation, but could be improved. My understanding is that there are three elements to the testing, covered in Sect. 3.3:

1.  Tests in which NO2 and CO fields generated by varying the scale factors in Eq (5) are input to the optimization algorithm and it has to reproduce the scale factors used.

2.  Comparing the NO2 and CO line densities and their ratio produced by the optimization against those from TROPOMI, in Fig. 4 and 5.

3.  Comparing the OH concentrations, NOx emissions, and NOx lifetimes output by the optimization to those derived from the EMG method (Table 2)

These are important tests, but each have weaknesses. Sources

- For #1, because the framework that generated the synthetic NO2 and CO fields is the same framework used to match them, it cannot account for chemistry or other confounding factors outside that framework.

- For #2, the optimization was given the goal of matching the TROPOMI NO2 & CO values and their ratio. Thus, showing that it can do so proves that the optimization has sufficient degrees of freedom and that the underlying model simulations include enough of the physics to reproduce the observations. It does not necessarily show that it obtains the right answer for OH.

- For #3, the EMG method makes a similar assumption to the optimization approach that the OH + NO2 pathway dominates NOx loss. This may well be true in Riyadh, but cannot give any information on errors from unsimulated chemistry.

One way to address these issues would be to repeat the first experiment, but using NO2, CO, and OH from a full chemistry simulation of WRF-Chem. Even if computational costs limit the runs to only a few days each in the summer and winter, comparing the OH returned by applying this optimization approach to the NO2 and CO columns simulated in the full chemistry WRF-Chem to the OH in that WRF-Chem run would be a useful metric of the error introduced from ignoring other NOx loss processes. Additionally, going back to my second main suggestion, this would be a useful way to demonstrate that this optimization approach works for individual days.

Since the authors state that full chemistry simulations are beyond the scope of this paper, I accept that this specific approach may not be practical. However, something like this - effectively an OSSE experiment in which NO2 and CO columns simulated with more complete chemistry are ingested by the optimization framework proposed in this paper, and the optimum OH from this framework compared with known OH in the original simulation - would help quantify the uncertainty introduced by omitting VOC-NOx chemistry from this framework.

Author Response:

*To strengthen the link between the TROPOMI observed $NO_2$ decay and OH we have decided to follow the suggestion made by the reviewer to extend our OSSE tests with an additional experiment in which the VOC-NOx chemistry in WRF-chem is used to test if OH can be recovered using our method. Section 3.3 has been extended with:*

*To obtain a more realistic estimate of the uncertainty in least squares optimization derived OH, TROPOMI data have been replaced by $NO_2$, CO and $NO_2$/CO ratio derived from WRF-chem using the Carbon Bond Mechanism Z (CBM-Z) gas-phase chemical mechanism (Zaveri and Peters, 1999). EDGAR based VOCs, NOx and CO emission have been used in combination with boundary condition for NO, $NO_2$, CO, ozone ($O_3$) from CAMS to run WRF-chem for August 17th, 2018 and November 18th, 2018 representing a summer and winter day, respectively.*

*For August 17$^{th}$, 2018, the ratio and XNO$_2$ optimization increase the CAMS based prior OH of 1.19x10$^7$ molecules/cm$^3$ by 15.7 % and 13.4 %, respectively (see Figure S17). In the WRF-chem full online chemistry simulation the boundary layer averaged OH for the box of 300 km x 100 km amounts to 1.33x10$^7$ molecules/cm$^3$, which <5 % lower than the optimized OH value that is derived using our method. The optimized NOx and CO emission differs by <11% than the emission input in full online chemistry. In winter, optimization increases CAMS based OH of 1.03x10$^7$ molecules/cm$^3$ by 19.4%. The OH derived from WRF-chem full online chemistry is 1.07x10$^7$ molecules/cm$^3$ and lower by 15.2% than the optimized OH value. The component wise optimization increases the EDGAR NOx and CO emissions by 23.1 % and 10.5 %, respectively. Overall, the uncertainty in optimized NOx, CO emission and OH derived from this test is <11 % in summer and 10 % to 23 % in winter. Since the lifetime of NOx is determined by other reactions in addition to the oxidation to HNO$_3$ considered in our method, it is expected to overestimate the real OH value. The test using WRF full chemistry confirms that this is indeed the case.*

[Figure]

Figure S17. WRF derived a) XNO$_2$/XCO, b) XNO$_2$ and c) XCO before and after optimization in comparison to WRF using full chemistry with CBMZ chemical scheme for 17$^{th}$ August, 2018.

[Figure]

[Figure]

[Figure]

Figure S18. Same as Figure S17 but for 18th November, 2018.

*Furthermore, to check if the size of the error matches the expected contribution of other NOx removal pathways the Chemistry Land-surface Atmosphere Soil Slab (CLASS) (van Stratum et al., 2012) model has been used. CLASS provides Ox-NOx-VOC-HOx photochemistry scheme with 28 different chemical reaction including the loss of $NO_x$ via $N_2O_5$ to $HNO_3$. We run the CLASS model for a summer and winter day representative of Riyadh. During the summer mid-day, $NO_x$ loss is dominated by OH (93.4 %) in CLASS. The heterogeneous $N_2O_5$ loss accounts for 6.6 % (see Figure S26), in close agreement with the full chemistry WRF test. During the winter mid-day, the $N_2O_5$ loss increases to 21.4 % and $NO_2$+OH accounts for 78.6 % of the total $NO_x$ loss (see Fig S26), which is larger than the mismatch in the full chemistry test, but within its uncertainty.*

[Figure]

Figure S26. The different pathways of NOx loss over Riyadh at the time TROPOMI overpasses during summer (left) and winter (right) , 2018.

**Minor comments**

- Title should be "Estimation of OH in an urban plume" or "Estimation of OH in urban plumes" (singular/plural mismatch in the current title) (done)

  Author Response

  *-Changed as suggested*

- Recommend defining XNO2 in the abstract, since it is less common to use column-average mole fractions for NO2 than for CO or CO2.

  Author Response :

  *In line 12 to 14, this sentence has been added: "A new method is presented for estimating urban hydroxyl radical (OH) concentrations using the downwind decay of Tropospheric Monitoring Instrument (TROPOMI) derived dry column mixing ratios of nitrogen dioxide ($XNO_2$)/carbon monoxide (XCO) ratios combined with Weather Research Forecast (WRF) model simulations."*

- At line 49, recommend mentioning that the EMG method assumes that $OH+NO_2$ is the only loss

  route so that this is clear from the start.

  Author Response :

  *In line 57 to 58, this sentence has been added: "In the EMG method, the satellite observed exponential decay of $NO_2$ downwind of the city centre is used to quantify the first order loss of $NO_2$, which is used to quantify the hydroxyl radical (OH) neglecting other NOx removal pathways."*

  In Sect. 2.6, do you use the average pressure and temperature over the same time period as the EMG fit when computing the rate constant? Over what vertical distance?

  Author Response:

  *In line 264, this sentence has been added: "Rate constants at the time of TROPOMI overpasses are obtained from WRF by averaging the IUPAC second order rate constant from the surface to top of the planetary boundary layer."*

- Recommend reiterating that f_emis, f_OH and f_Bg in Eq. (5) are the scale factors alongside the other variable descriptions following Eq. (10). Also please explain why they are divided by 10.

Author Response

*This sentence has been added in line 293 to 294 "The scaling factors $f_{emis}$, $f_{OH}$ and $f_{Bg}$ obtained from the ratio optimization have been multiplied by 10 % (i.e. divided by 10) as they represent changes in emission ratio, OH and Bg by 10 %."*

*This sentence has been added in line 308 to 309 "The scaling factors $f_{emis}$, $f_{OH}$ and $f_{Bg}$ are divided by a factor 10, because $\Delta XNO_{2\,emis}$, $\Delta XNO_{2\,OH}$ and $\Delta XNO_{2\,Bg}$ are defined as 10 % changes in NOx emission, OH and background level. "*

- Fig. 3 caption - I'm not sure "zonally" is the right term, that implies averaging along latitude lines. Should this be perpendicular to the wind direction?

Author Response :

*The caption of Figure 3 has been changed to: "Comparison of WRF and TROPOMI averaged across the wind direction for a) $XNO_2$, b) XCO and c) WRF Ratio ($XNO_2$/ XCO) without CAMS background d) WRF Ratio ($XNO_2$/ XCO) with background and TROPOMI as a function of distance to the center of Riyadh for summer (June, 2018 to October, 2018). "*

- Fig. 3 caption - "with background as a function of distance" is ambiguous - does it mean that the background value depends on distance or is it saying that each of the quantities described previously (XNO2, XCO, WRF Ratios) are plotted versus distance.

Author Response:

*This comment is accounted for also by the new formulation of the figure 3 caption.*

- Line 357 - the OH uncertainties of 11% to 15% are probably underestimated because VOC chemistry is not accounted for. Please note that here.

Author Response:

*The OH uncertainties arising from different NOx removal pathways have been included in section 3.4. The sentence is added in Line 400 "For summer and winter, the uncertainties of the optimized OH concentrations is <17 % and < 29 % respectively. For NOx and CO emissions, the uncertainty is < 29 % in summer and winter."*

- Lines 358 to 368 - the discussion here is difficult to follow because the results for OH, emission ratio, and background ratio vs. CAMS are very spread out and (in one case) given in different ways, e.g. the amount by which CAMS is overestimated and optimized value. It would help significantly to gather the results from the ratio-optimized and component-optimized tests into a table along with the CAMS values and provide the actual values. Describing the optimization results as percentages by which CAMS is overestimated is awkward to follow while reading.
- Author Response:

*Table S4 has been added to summarize the prior and optimized emission ratios, OH, and background ratios obtained from the ratio and component wise optimizations.*

**Table S4. Overview of optimized emission ratio, OH and background ratio using ratio and component wise optimization.**

| Variables | Summer | | | Winter | | |
|---|---|---|---|---|---|---|
| | Prior | Ratio optimization | Component optimization | Prior | Ratio optimization | Component optimization |
| Emission ratio ($NO_x/CO$) | 0.79 (8.2/10.34) | 2.01±0.33 | 0.55±0.091 (11.6/21.09) | 0.93 (9.4/10.1) | 1.46±0.8 | 0.36±0.18 (7.8/21.6) |
| OH ($10^7$, molecules/cm$^3$) | 1.3 | 1.7± 0.32 | 1.66±0.29 | 0.86 | 1.3±0.38 | 1.28±0.37 |
| Background ratio ($XNO_{2Bg}/XCO_{Bg}$) | 0.002 (0.22/92.13) | 0.00068± 6.12e-05 | 0.00059 (0.053/88.41) | 0.0016 (0.15/92.58) | 0.00053 ±0.00015 | 0.00054±0.00015 (0.049/90.54) |

Lines 365 to 371 - the discussion of why the component and ratio optimizations yield different emissions ratios isn't convincing. Whether directly optimizing the ratio or the NOx and CO amounts, the algorithm has information on the ratio of NO2 to CO, so how can it come up with emissions ratios that vary from 0.38 to 1.05 (if I understood the ratio optimization result correctly)? If the component optimization matches TROPOMI NO2 and CO well, it should by definition match the TROPOMI NO2/CO ratio too.

Author Response :

*The ratio inversion has enough degrees of freedom to get the observed $NO_2/CO$ ratio correct. However, as different emissions can have the same emission ratio, the degree of freedom of the absolute emissions is larger as was explained in the manuscript already. It puzzled us also why even the emission ratio could be different between the ratio and component wise optimization. It turned out the emission ratio is sensitive to the background. This can be understood considering that the $NO_2/CO$ ratio over a city is the sum of the contributions of the background and the city emission. The relative weight of the two is determined by the absolute background levels and absolute emissions of CO and $NO_2$. In the ratio inversion these absolute values are not well constrained, as the data only inform about ratios. This is our explanation of why the optimized solutions can have different emission ratios. According to our tests, however, the OH concentration derived from our method is not affected as it is not sensitive to the background.*

- Lines 371 to 376 - please provide the Lama et al. (2020) values for comparison.

Author Response :

*In line 405, this sentence has been added "Lama et al., (2020) inferred an $NO_2/CO$ emission ratio over Riyadh of 0.47 ± 0.1 for 2018 from TROPOMI favoring the Monitoring Atmospheric Chemistry and Climate and CityZen (MACCity) emission ratio over that of EDGAR."*

- Lines 391 to 392 - "Both methods result in higher NOx emissions and shorter lifetimes in summer; lower NOx emissions and longer lifetimes in winter." In summer, the prior values are within the EMG uncertainties. To claim that the EMG values are higher than the prior, given the uncertainty range, requires a t-test or other statistical test to determine if that difference is significant.

Author Response :

*Here, the higher and lower values are not meant relative to the prior, but relative to each other as a quantification the seasonal changes which happen to be consistent between the two methods and in line with the expected seasonal differences.*

- Line 411 - the simplified OH + NO2 chemistry used here will also be a barrier to more general use. It would be good to acknowledge that any such simplified approach in the future will either need to (a) account for other paths for NOx loss, or (b) prove that neglecting those paths introduces an acceptable error in the OH concentrations.

Author Response

*In line 459 to 461, the sentence has been added "In the future, the accuracy of our method can be further improved by accounting other NOx removal pathways. "*

- Line 442 - this paragraph could use a stronger topic sentence. It's not clear what the main point of this paragraph is.

Author Response

*The paragraph has been removed.*

---

## Author Response (AR2)

I want to thank the authors for their significant effort to address the comments of both reviewers in this revision. The changes have significantly improved the manuscript, both in content and organization. There are a few aspects of the original comments which could be more fully accounted for, and a few minor issues arising in the revisions themselves. Once these are addressed, I recommend publication.

Notes on responses to reviewer 1
* * *
1. Re: the comment "In terms of providing context/motivation for this work, I would [be] interested in seeing discusson on topics such as..."

The sentences added by the authors are a good start in addressing this comment, but could go farther in addressing the broader impact of the work. The added text says that "The TROPOMI retrieved XNO2/XCO ratio is useful for estimating mid-day OH over isolated localized sources..." but does not explicitly tie that to a larger science question or societal benefit. An additional sentence or two explaining what scientific question(s) this could help answer (could it provide information on the dominant NOx chemical regime for example?) or how it will help e.g. improve future air quality would more completely address the reviewer's comment.

**Author Response:**

*This sentence is added to the discussion section:*
*"The TROPOMI retrieved $XNO_2$/XCO ratio is useful for estimating mid-day OH over isolated localized sources, such as the city of Riyadh, showing a clear contrast between the urban plume and the background. Such TROPOMI derived OH estimates offer a new opportunity to evaluate urban photochemistry in chemistry transport models. OH depends non-linearly on NOx and VOC emission, meteorological conditions, etc. (Sillman et al., 1990), which vary substantially between cities that are monitored by TROPOMI. Therefore, the application of our method to the global and multi-year dataset that is available could contribute substantially to the understanding of urban photochemistry and the development of effective pollution mitigation strategies. "*

Additionally, the discussion of what cases this method could be applied to is fairly abstract. One of reviewer #2's comments pointed out that Riyadh is an ideal case for this technique because it is a large, isolated point source with minimal clouds, and suggested testing the percentage of clear sky pixels required by withholding increasing percentages of data and seeing how the answer changed. If done with the OSSE study added in section 3.3, then the authors could quantify the change error with decreasing data availability. I strongly recommend including this test, which would help quantity the statement "isolated tropical and subtropical cities are best suited for this method", as cities in tropical Amazonia (for example) will frequently lose significant amounts of data due to clouds.

**Author Response:**

*As suggested by the reviewer, we applied a bootstrapping method to the data of 17th August and 18th November, 2018 to quantify the impact of a reduced data availability on the estimation uncertainty for OH, NOx and CO emissions. 500 images were generated repetitively while randomly reducing the TROPOMI data size (N) to 50 % of the original datasets ($N_R$). For each image, the OH concentration, NOx and CO emissions were estimated using our least square optimization method.*

*From the ensemble statistics we infer that the uncertainty of optimized OH is <15 % for summer and <21 % in winter due to the reduced data availability, which is smaller than the total estimation error (listed in Text S6, Tables S1 and S2). Similarly, for NOx and CO emissions, the uncertainty is < 15 % for summer and winter.However, this outcome depends on the random sampling process in the boot strapping method. In reality, the data gaps due to cloud cover are distributed differently. Since we select cloud free days, the bootstrapping test is not representative for our method and was therefore not included in the paper. A tropical case, cloud free days would obviously be less abundant than over Riyadh. However, we have shown that a single cloud free day may suffice to obtain a useful OH estimate.*

2. Re: the comments about adding statements on the iterative optimization and CAMS use to the abstract

The authors have addressed these comments adequately, however the abstract has grown quite long and dense. I might suggest that the authors (a) make the results their own paragraph, and (b) reduce the number of results presented in the abstract to focus on the 1 or 2 most important ones.

**Author Response:**

*We condensed the abstract as follow:*

*A new method is presented for estimating urban hydroxyl radical (OH) concentrations using the downwind decay of the ratio of nitrogen dioxide over carbon monoxide column mixing ratios ($XNO_2 / XCO$) retrieved from the Tropospheric Monitoring Instrument (TROPOMI). The method makes use of plumes simulated by the Weather Research and Forecast model (WRF-CHEM) using passive tracer transport, instead of the encoded chemistry, in combination with auxiliary input variables such as Copernicus Atmospheric Monitoring Service (CAMS) OH, Emission Database for Global Atmospheric Research v4.3.2 (EDGAR) NOx and CO emissions, and National Center for Environmental Protection (NCEP) based meteorological data. $NO_2$ and CO mixing ratios from the CAMS reanalysis are used as initial and lateral boundary conditions. WRF overestimates $NO_2$ plumes close to the center of the city by 15 % to 30 % in summer and 40 % to 50 % in winter compared to TROPOMI observations over Riyadh. WRF simulated CO plumes differ by 10 % with TROPOMI in both seasons. The differences between WRF and TROPOMI are used to optimize the OH concentration, NOx, CO emissions and their backgrounds using a iterative least square method. To estimate OH, WRF is optimized using a) TROPOMI $XNO_2/XCO$, b) TROPOMI derived $XNO_2$ only.*

*For summer, both the $NO_2/CO$ ratio optimization and the $XNO_2$ optimization increase the OH prior from CAMS by 32 ± 5.3 % and 28.3 ± 3.9 % respectively. EDGAR NOx and CO emissions over Riyadh are increased by 42.1 ± 8.4 % and 101 ± 21%, respectively, in summer. In winter, the optimization method doubles the CO emissions also, while increasing OH by ~52 ± 14 % and reducing NOx emission by 15.5 ± 4.1 %. TROPOMI derived OH concentrations and pre-existing Exponentially Modified Gaussian function fit (EMG) method differ by 10 % in summer and winter, confirming that urban OH concentrations can be reliably estimated using the TROPOMI-observed $NO_2/CO$ ratio. WRF optimization method can be applied to single TROPOMI overpass, allowing to analysis day to day variability in OH, NOx and CO emission.*

3. Re: the comment about v2.2.0 of TROPOMI data

Thank you to the authors for going back and incorporating the newer data version at least in the error calculation. Two notes:

- Please include a reference to Text S6 in the captions for tables S1-S3 so that the reader can find the methodology for the error calculation more easily.

**Author Response:**

*Changed as suggested*

- One potential issue in using v1.2.x and v1.3.x of TROPOMI data in the main analysis but v2.3.x in the error analysis is if any of the changes between the two TROPOMI data versions affect the tropospheric slant column. Based on my reading of sect. 5 of the NO2 readme (https://sentinels.copernicus.eu/documents/247904/0/Sentinel-5P-Nitrogen-Dioxide-Level-2-Product-Readme-File/3dc74cec-c5aa-40cf-b296-59a0f2140aaf), that is not the case - it looks like most changes affect the AMF or the stratospheric column. However, it would be good to acknowledge this inconsistency in Text S6 and include links to the NO2 readme and the PAL page (https://data-portal.s5p-pal.com/product-docs/no2/PAL_reprocessing_NO2_v02.03.01_20211215.pdf) so readers can understand the differences themselves.

*In our opinion the use of v2.3.x does not introduce an inconsistency as we only use the difference with v1.2.x and v1.3.x to obtain an, admittedly crude, estimate of systematic error. We agree, however, to specify the differences between these retrieval versions using the suggested references.*

4. Re: the comment "The authors assessed NO2 data quality vs. ground-based measurements...is there a similar analysis that can be done for CO?"

I concur with the authors' explanation in the response. I'd suggest adding something to the end of Sect. 2.2 that points the reader to Text S6 where this is discussed.

**Author Response:**

**-** *In Line 131, this sentence is added: "The comparison of TROPOMI derived XCO to the 28 different TCCON ground based station suggest that difference between TCCON and TROPOMI is in the range of 9.1 ± 3.3 % (Shah et al., 2020). Such difference is used to estimate the uncertainty in the NOx emission and life time (see Table S1, S2, S3 and Text S6). "*

5. Re: the comment "I'm not sure I agree with this justification for not allowing XNOx,Bg to be lost by OH..."

Here, I don't follow the authors' justification that because WRF and CAMS are within 10-20% this means that the background NOx is in photochemical equilibrium. I'm not sure from this response whether the comparison is WRF vs. CAMS or summer vs. winter. It's also not clear to me whether the "WRF" in this response is the passive tracer simulation or the full chemistry test (though only the latter would make sense to me). If the statement "The application of OH to the NOx background results in much smaller background NOx concentrations in WRF than in CAMS" means that the authors' applied the CAMS OH concentrations as fixed data to the passive tracer WRF background NOx and compared it to colocated NOx from a full-chemistry CAMS simulation, I'm not surprised that there is such a large difference as the fixed OH fields in this case wouldn't respond to the changing NOx-HOx equilibrium in the WRF simulation.

But, I think Table R1 communicates the point needed - that even if OH is treated as a fixed concentration and applied to the background NOx tracer, the difference in both derived NOx emissions and OH concentrations is reasonably small. (I am surprised that NOx emissions decrease in the summer with Bg OH-NOx loss test - I would have expected lower background to need greater emissions to make up the plume magnitude. But the w/BG OH NOx emissions value is within the uncertainty of the no BG OH NOx value.) I would recommend the authors' include Table R1 in the supplement and reference it in the discussion of the treatment of background NOx.
**Author Response:**

*This sentence is added in the section 4, line 525 to 527: "A sensitivity test has been performed in which $XNO_{x,Bg}$ is lost by OH. In this case the optimized NOx emission and OH for summer and winter differ by < 7 % from the default method where the background is treated as an inert tracer (see Table S6)."*

6. Re: the comment "Please state why this model simulation is well suited to evaluate emission changes..."

I'm unclear on Fig. R3 - in the right panel, is XCO_WRF_opt the optimization result using the prior reduced by the factor of 10? If so, this is a nice sensitivity test and should go into the supplement (with the meaning of the plot series in the right panel clarified), not just the response.

**Author Response:**

*This sentence is added to the discussion section: "Furthermore, a sensitivity test has been performed in which the prior emission has been changed. The optimized emission varied by < 5 %, demonstrating robustness of the method to the choice of prior (see Fig S24). This also indicates that the optimization method can be used to study emission changes. "*

7. Re: comments on various URLS:

- The https://cophub.copernicus.eu/s5pexp link in the data availablility statement still gives a "Not found" error
   **Author Response:**

*The link has been modified to " https://s5phub.copernicus.eu ; http://www.tropomi.eu "*

- The Zenodo link given in the data availability statement \*does\* require a login, this is just a link to your personal account's list of deposited data. If I follow it and log in, it shows my data, not yours. Please provide the DOIs or DOI URLs given in the Zenodo page for each dataset (and please test these links in a private browser window - that is the best way to ensure they are truly public).
**Author Response:**

*The Zenodo link is as follows: " https://doi.org/10.5281/zenodo.5752219 ". To make sure the link is public, I sent it to 10 different persons who had no problem accessing the data without login.*

Notes on response to reviewer 2
* * *
1. Re: the comment "The larger issue is the choice to use passive tracers..."

Thank you to the authors for clarifying that this was to reduce the computational cost. Two new comments on the added sentences:

- It's not clear how this method helps separate out the effect of meteorology, OH concentration, NO2 + OH rate constant, and NO2/CO ratio. The OH concentration and NO2/CO ratio is clear, but given that the model meteorology and NO2 + OH rate constants will still have errors and there is no optimization of wind direction or kinetics, I don't think you can say that this method isolates the effect of meteorology and kinetics.

**Author Response:**

*We agree with the reviewer that transport model uncertainties influence the comparison with satellite data in a way that is difficult to quantify. However, as TROPOMI measures the $NO_2$ and CO from same platform and taking their ratio reduces the impact of transport model uncertainties. The model allows to compute these influences on the $NO_2$/CO ratio. There is no way our optimization method can separate the uncertainty in OH from uncertainties in reaction kinetics, but we do not claim this.*

*In Section 3.2, line 354 this explanation is provided as follow: " As expected, $Ratio_{without\ OH}$ shows an approximately straight line when the background is removed, because transport influences $NO_2$ and CO in the same way and therefore cancels out in the ratio (see Fig. 3b). The $Ratio_{with\ OH}$ however, shows an approximately gaussian relation with distance due to the influence of the sink on $NO_2$. This comparison demonstrates the sensitivity of the relation between $XNO_2$/XCO ratio and downwind distance to the $NO_2$ lifetime, which we want to exploit to quantify OH. "*

- The statement that this method is an important improvement over the EMG method because it includes the use of a transport model isn't quantified. Introducing a transport model isn't an automatic improvement if it doesn't result in better estimates of OH, better estimates of emissions, or some other quantifiable improvement. Please be specific and quantitative about how it is an improvements - either show with the new simulations in Sect. 3.3 that the WRF optimization produces more accurate OH or emissions, or focus on how this method enables analysis of day-by-day OH and emissions, whereas the EMG method needs longer time periods.

**Author Response:**

*The updated version of the manuscript includes this explanation (section 3.6, line 476): "In contrast to EMG method, the optimization method can be used for a single TROPOMI overpass (see Section 3.6) and does not require yearly averaged $NO_2$ data, allowing analysis of day-by-day OH, NOx and CO emission (see Section 3.3). Segregation and averaging of $NO_2$ urban plume by wind sector is not required in the optimization method. "*
*In the conclusion section (line 567) we explain the difference as follow: "With our method, single TROPOMI overpasses can be used to estimate OH whereas the EMG method requires averaging of urban $NO_2$ plumes by wind sector.*

2. Re: the comment about showing that this method works for single-day overpasses

Thank you for including a single day results. In combination with the new Figs. S17 and S18, this is very promising. Two notes:

- Why in the new text do you use r2 for the initial WRF XNO2 and XCO comparisons, but X2 for the optimized comparisons? It would be nice to use X2 for both to be consistent.

**Author Response:**

*r2 and X2 quantify something different and their use serves different purposes. r2 is used as measure of spatial correlation, whereas X2 is used to test the goodness of the optimized fit. Since these numbers were not meant to be compared, we do not see the need to make them consistent.*

- The original reviewer comment also mentioned testing what percentage of data must be cloud free for this method to work by essentially bootstrapping smaller and smaller percentages of data and checking the results. This would work especially well with the new synthetic tests in Sect. 3.3, and having a quantification of how much clear sky data is necessary would strengthen the discussion in Sect. 4 about how widely this method could be used. There the authors suggest that this method will work best for tropical and subtropical cities, but cities in e.g. the Amazonian tropics will often have a high fraction of cloudy pixels. Could such a bootstrapping analysis be added to the supplement to support Sect. 4?

**Author Response:**

*This question is addressed in the answer to question 1 of reviewer 1.*

Other notes on revised text
* * *
Line 12: why are emissions not mentioned along with OH concentrations as an output of this method?

**Author Response:**

*The main objective of this paper is to estimate OH. As suggested by reviewer 1, we tried to reduce the size of the abstract, prioritizing the main objective. For this reason, we decided to keep it as is.*

- Line 113: there are numerous citations you could use to support the statement that high resolution NO2 priors better resolve the NO2 gradients, to name a few:
* www.atmos-chem-phys.net/11/8543/2011/
* https://acp.copernicus.org/articles/14/3637/2014/
* https://acp.copernicus.org/articles/15/5627/2015/

**Author Response:**

*The citations are added in line 113.*

- Line 271: is the PBL height here taken from the WRF simulations?

**Author Response:**

*In line 271, we added: "The PBL height at the time TROPOMI overpass has been taken from WRF"*

Line 311: could clarify the interpretation of f_emis, f_OH, f_Bg by specifying whether f = 0 or f = 1 means that the prior was correct. Normally, I think of "scale factors" as being multiplied by the prior (so f = 1 would mean the prior was correct) but here it looks like f = 0 means that instead.

**Author Response:**

*The scaling factor $f_{emis}$, $f_{oh}$ and $f_{bg}$ represent the modification of the prior in percentage change. The scaling factors $f_{emis}$, $f_{OH}$ and $f_{Bg}$ obtained from the ratio optimization have been divided by 10*

*because* $\frac{\Delta F}{\Delta emis}$ , $\frac{\Delta F}{\Delta OH}$ *and* $\frac{\Delta F}{\Delta Bg}$ *are defined as the change in F due to modification of emission, OH and background by 10 %.*

Line 463: what does "For winter, the dissimilarity between the EMG method and the prior reduces after optimization," mean? Do you mean that the difference between the EMG and WRF-optimized results are smaller than the difference between the EMG results and the prior? As written, it sounds like both the EMG and prior are being optimized somehow.

**Author Response:**

*The sentence is modified to "For winter, the difference between the EMG and WRF-optimized results are smaller than the difference between the EMG results and the prior "*

Text S1: in step 2, please specify which reaction the second-order rate constant in question is for

**Author Response:**

*The sentence is modified into: "First, we calculate the IUPAC second order rate constant for the reaction between NO$_2$ and OH, using the pressure and temperature for each vertical level.*

- Table S4: please add to the caption what the quantities in parentheses are

**Author Response:**

*Changed as suggested.*